# Nano-Zinc Oxide Can Enhance the Tolerance of Apple Rootstock M9-T337 Seedlings to Saline Alkali Stress by Initiating a Variety of Physiological and Biochemical Pathways

**DOI:** 10.3390/plants14020233

**Published:** 2025-01-15

**Authors:** Jietao Zhai, Xulin Xian, Zhongxing Zhang, Yanxiu Wang

**Affiliations:** College of Horticulture, Gansu Agricultural University, Lanzhou 730070, China; 18093215893@163.com (J.Z.); 18394159118@163.com (X.X.); zhangzx123478@163.com (Z.Z.)

**Keywords:** ASA-GSH cycle, ion homeostasis, M9-T337, oxidative damage, saline alkaline stress, osmoregulation, ROS removal, hormones

## Abstract

Soil salinization severely restricts the growth and development of crops globally, especially in the northwest Loess Plateau, where apples constitute a pillar industry. Nanomaterials, leveraging their unique properties, can facilitate the transport of nutrients to crops, thereby enhancing plant growth and development under stress conditions. To investigate the effects of nano zinc oxide (ZnO NP) on the growth and physiological characteristics of apple self-rooted rootstock M9-T337 seedlings under saline alkali stress, one-year-old M9-T337 seedlings were used as experimental materials and ZnO NPs were used as donors for pot experiment. Six treatments were set up: CK (normal growth), SA (saline alkali stress,100 mmol/L NaCl + NaHCO_3_), T1 (saline alkali stress + 50 mg/L ZnO NPs), T2 (saline alkali stress + 100 mg/L ZnO NPs), T3 (saline alkali stress + 150 mg/L ZnO NPs) and T4 (saline alkali stress + 200 mg/L ZnO NPs). The results were found to show that saline alkali stress could significantly inhibit the growth and development of M9-T337 seedlings, reduce photosynthetic characteristics, and cause ion accumulation to trigger osmotic regulation system, endogenous hormone and antioxidant system imbalances. However, the biomass, plant height, stem diameter, total leaf area and leaf perimeter of M9-T337 seedlings were significantly increased after ZnO NP treatment. Specifically speaking, ZnO NPs can improve the photosynthetic capacity of M9-T337 by increasing the content of photosynthetic pigment, regulating photosynthetic intensity and chlorophyll fluorescence parameters. ZnO NPs can balance the osmotic adjustment system by increasing the contents of soluble protein (SP), soluble sugar (SS), proline (Pro) and starch, and can also enhance the activities of enzymatic (SOD, POD, and CAT) and non-enzymatic antioxidant enzymes (APX, AAO, GR, and MDHAR) to enhance the scavenging ability of reactive oxygen species (H_2_O_2_, O_2_^•−^), ultimately reducing oxidative damage; ZnO NPs promoted the growth of M9-T337 seedlings under saline alkali stress by synergistically responding to auxin (IAA), gibberellin (GA_3_), zeatin (ZT) and abscisic acid (ABA). Additionally, the Na^+^/K^+^ ratio was reduced by upregulating the expression of Na^+^ transporter genes (*MdCAX5*, *MdCHX15*, *MdSOS1*, and *MdALT1*) and downregulating the expression of K^+^ transporter genes (*MdSKOR* and *MdNHX4*). After comprehensive analysis of principal components and correlation, T3 (150 mg/L ZnO NPs) treatment possessed the best mitigation effect. In summary, 150 mg/L ZnO NPs(T3) can effectively maintain the hormone balance, osmotic balance and ion balance of plant cells by promoting the photosynthetic capacity of M9-T337 seedlings, and enhance the antioxidant defense mechanism, thereby improving the saline alkaline tolerance of M9-T337 seedlings.

## 1. Introduction

Soil salinization is a severe ecological and socio-economic issue faced globally, posing a significant barrier to the sustainable development of agricultural production [1]. Research indicates that approximately 7% of the global land area and 20% of arable land are affected by salinization [1]. In China, the area of land affected by salinization is particularly alarming, reaching 99 million hectares, primarily distributed in the northwest inland and eastern coastal regions [2]. Saline alkali stress can induce osmotic stress, ion toxicity, and oxidative stress, leading to a significant increase in the accumulation of reactive oxygen species (ROS) within cells. These combined effects can cause metabolic dysfunctions, including membrane damage, nutritional imbalances, enzyme inhibition, and impaired photosynthesis, ultimately leading to plant death [3]. Therefore, enhancing the saline alkali tolerance of apples is crucial for promoting the sustainable development of the apple industry.

During plant growth and evolution, salt-tolerant plants are apparently able to adapt to environmental changes through a complex set of molecular and genetic regulatory mechanisms. Specifically, with the ion regulatory pathway, plants rely primarily on ion transport proteins (Na^+^/H^+^ reverse transporter proteins and K^+^ channels) in the cell membrane to maintain the intracellular balance of Na^+^/K^+^ ratios, thereby preventing ion toxicity [4]. In addition, salt-tolerant plants effectively scavenge excess reactive oxygen species (ROS) by increasing the activity of antioxidant enzymes (SOD, POD, CAT) and accumulating non-enzymatic antioxidants (APX, AAO, GR, and MDHAR), thereby protecting cells from oxidative damage [5]. At the same time, plants increase cellular osmotic pressure and thus maintain the stability of cellular structure and water balance by accumulating osmoregulatory substances, such as Pro, SS and GB; phytohormones (including ABA, C2H4 and GA_3_) play a key role in regulating the plant’s adaptation to saline and alkaline stresses, and they can enhance the plant’s salinity tolerance by affecting the pathways of growth, stomatal opening and closing and the synthesis of osmoregulatory substances [6].

Salt tolerance is the result of synergistic expression of multiple genes involving multiple signaling pathways, including the abscisic acid pathway, the SOS pathway, and the protein kinase pathway [7]. Among these, in the SOS pathway, the SOS1, SOS2, and SOS3 genes act together to maintain intracellular ionic homeostasis by regulating Na^+^/K^+^ balance, thereby enhancing salt tolerance in plants [7]. In addition, transcription factors such as the *MYB* family regulate the synthesis of antioxidants and the transport of ions by modulating the expression of downstream genes, further improving the salt tolerance of plants [8]. In the abscisic acid pathway, ABA binds to the receptor and inhibits *PP2C*, activating the *SnRK*2 kinase, which in turn phosphorylates the transcription factors (such as bZIP), and regulates the expression of salt tolerance genes [8]. Notably, recent studies have revealed that the receptor-like kinase FERONIA affects plant salt tolerance by regulating the metabolic flow of the photorespiratory pathway, providing a new perspective for understanding the mechanism of plant salt tolerance [9]. These complex regulatory mechanisms interact with each other and together enable salt-tolerant plants to survive and grow under saline stress.

With the advancement of science and technology and the accumulation of nanotechnology, the application prospects of nanomaterials in agriculture are being continuously explored. Nanoparticles (NPs) are widely used in the agricultural field, and have a positive effect in terms of improving plant disease resistance, nutrient absorption efficiency and yield [10,11,12]. Studies have found that leaf spraying or root application of nanoparticles (NPs) can be absorbed and utilized by the plant, and ultimately distributed in all organs of the plant [13,14]. The effects of nanomaterials on plant growth and development are primarily manifested in enhancing seed vigor, increasing the activity of various enzymes involved in metabolic pathways, promoting plant metabolism, improving the use efficiency of water and nutrient by plants, enhancing the plant’s resistance to biotic and abiotic stresses, and ultimately increasing yield and improving quality [15,16].

As the third most commonly used metal nanomaterial, nano-zinc oxide (ZnO NPs) has the advantages of non-toxicity, high surface energy, high catalytic efficiency, mobility and fluorescence, advantages which make it superior to ordinary zinc oxide [17]. Further studies have also found that nano zinc oxide has a wide range of adaptability, it is not only suitable for crops, but can also be used for pasture, horticultural plants and many other types of plants, this wide range of applicability means that the use of nano zinc oxide in the field of plant protection has a broad range of application prospects and can be used to provide effective protection measures for different types of plants and agricultural production. In the field of agriculture, the soil application of nano-zinc oxide has a slower and better effect than foliar application, because it can stimulate the target nutrient to enter the plant system and because it has a higher nutrient use efficiency [18]. Recent research progress has showed that nano-zinc oxide can play a positive impact on crops. For example, exogenous application of ZnO NPs can improve the photosynthetic efficiency and antioxidant capacity of cabbage (*Brassica juncea*) [19], enhance the seed germination, plant length, and root development of tomato (*Solanum lycopersicum*) [20], improve the microstructure of rapeseed (*Gossypium hirsutum*) seeds, reduce the oxidative stress response of plants [21], and enhance the cadmium absorption and remediation capabilities in eggplants (*Solanum melongena*) [22]. In addition, the potential benefits of ZnO NPs in alleviating saline alkali stress have been confirmed in a variety of plants, such as *Lycium barbarum* (*Trigonella foenum-graecum*) [23], tomato (*Solanum lycopersicum*) [20], pea (*Vicia faba*) [24], rice (*Oryza sativa*) [25], cabbage (*Brassica oleracea*) [19], and sorghum (*Sorghum bicolor*) [26]. The study of Türkoğlu et al. further clarified that ZnO NPs have a significant effect on the synthesis of plant secondary metabolic pigments and the content of proteins and sugars in nutrient transport and can effectively reduce reactive oxygen radicals produced under plant stress [27]. Specifically, spraying ZnO NPs on safflower leaves can increase the percentage of water content, osmotic level and yield of safflower under saline alkali stress, significantly improve the antioxidant capacity and basic agronomic index parameters of safflower, and alleviate the damage of safflower under salt stress [28]. At the same time, Lalarukh et al. found that foliar application of ZnO NPs promoted the formation of osmotic adjustment substances and nutrient absorption in wheat under salt stress, promoted plant metabolism, and improved salt tolerance [29]. Azmat et al. discovered that treatment with ZnO NPs promotes the growth of wheat under high temperature and combined saline alkali stress by increasing biomass, photosynthetic pigments, nutrients, soluble sugar, protein and indoleacetic acid content, thereby enhancing the resistance of wheat to high temperature and combined saline alkali stress [30]. Dang et al. have also demonstrated that ZnO NPs can elevate the levels of glycine betaine (GB) and total soluble protein (TSP) in rice, enhance the activity of ribulose diphosphate carboxylase oxygenase, the light energy capture efficiency of photosystem II, and the ability of electron transport under salt stress [31]. Furthermore, under saline alkaline stress conditions, appropriate concentrations of ZnO NPs can activate the expression of genes associated with the accumulation and defense of reactive oxygen species (ROS), including AREB, NCED3 and CRK1, which helps to enhance seed germination capacity, root development and dry matter accumulation and to reduce chlorophyll decomposition and oxidative damage, while significantly improving photosynthetic efficiency and the activity of antioxidant enzymes [32].

M9, as one of the core parents with the most rootstocks, is a dwarf rootstock originating from Europe and widely used in the world. Among M9 rootstocks, the M9-T337 self-rooted rootstock exhibits desirable nursery traits, early fruiting, high yield, easy mechanized management and other characteristics and possesses a strong dwarfing effect, ensuring effectiveness while controlling the transport of nutrients effectively [33]. In recent years, the use of nano-zinc oxide has seen important progress in terms of promoting a variety of non-woody plants, though research on its application in forestry fruit tree crops is scarce [33]. Therefore, the apple self-rooted rootstock M9-T337 was used as the experimental material and ZnO NPs as the donor to investigate the growth, photosynthetic physiology, osmotic regulatory substances, and antioxidant enzyme systems of apple seedlings, in order to provide an effective strategy for the salt-tolerant cultivation of M9-T337 seedlings.

## 2. Materials and Methods

### 2.1. Test Materials

ZnO NPs were purchased from Jiangsu Xianfeng Nanomaterials Technology Co., Ltd. (Nanjing, China)., with a purity of 99 wt %. The nano-zinc oxide suspension was first stirred with a magnetic stirrer for 30 min, and then ultrasonically centrifuged for 30 min. Apple rootstock M9-T337 (*Malus domestica* Borkh.) was purchased from Shandong Huinong Horticultural Company Co., Ltd. (Linyi, China).

### 2.2. Treatment and Experimental Design

In April 2023, the apple rootstock M9-T337 seedlings were transplanted into a ceramic flowerpot (diameter 25 cm, depth 38 cm). The substrate was composed of 20% vermiculite, 20% perlite and 60% peat soil, with one seedling per pot. The experiment was carried out in the rain shelter of Gansu Agricultural University (N 36°1′–37°9′, E 106°21′–107°44′). The average temperature was 26 °C and the relative humidity was 50%. When the plants were cultivated to the seven–eight leaf stage, 270 seedlings with no pests and diseases and uniform growth were randomly selected. After ZnO NPs were dissolved by ultrasonic treatment, M9-T337 seedlings were uniformly irrigated in the form of suspension. Saline alkali treatment was performed on the seventh week after treatment. The following six treatments were set up: CK (normal growth), SA (saline alkali stress, 100 mmol/L NaCl + NaHCO_3_), T1 (saline alkali stress + 50 mg/L ZnO NPs), T2 (saline alkali stress + 100 mg/L ZnO NPs), T3 (saline alkali stress + 150 mg/L ZnO NPs), and T4 (saline alkali stress + 200 mg/L ZnO NPs), with 15 plants per treatment and 3 replicates. Among these, CK was watered with Hoagland nutrient solution. First, different concentrations of ZnO NPs were pretreated. With the exception of CK and SA, T1–T4 treatments were watered equally with 200 mL of ZnO NP suspension with different concentrations; this was undertaken five times, once every three days. After the last ZnO NP treatment had been undertaken after seven days, the saline alkali stress treatment was carried out; except CK, 200 mL saline alkali solution (100 mmol/L NaCl + NaHCO_3_) was poured into SA–T4 treatments three times and every seven days.

### 2.3. Measuring Indicators and Methods

#### 2.3.1. Characterization of ZnO NPs

The morphology and structural characteristics of ZnO NP were observed by transmission electron microscopy (Tem) (FEI G2F30, ThermoFisher, Waltham, MA, USA).

#### 2.3.2. Growth Parameter

One seedling of each treatment was randomly selected and placed in a black cloth background environment for photo observation. Tree seedlings of each treatment were randomly selected for the determination of plant height, stem diameter, leaf area, leaf circumference, fresh weight and dry weight.

Three seedlings were randomly selected from each treatment and the roots were completely cut and washed with UP water. The roots were scanned in a tray using a root scanner (Regent Corporation, STD-4800, Richmond, BC, Canada). Subsequently, WinRHIZO5.0 software (Regent, Hydro Quebec, Montreal, QC, Canada) was used to analyze the obtained root images to determine various indicators, including length, total root surface area, average root diameter, total root volume and number of root tips. Each treatment was repeated 3 times.

A weight of 0.5 g of the root tip tissue was weighed to assess for root activity using the triphenyltetrazolium chloride (TTC) method by Arnon et al. [34], Each treatment was repeated 3 times.

#### 2.3.3. Stomatal Morphological Structure

Following the method of Liu et al., the samples were processed and the stomata were observed and photographed using a scanning electron microscope (SEM S-3400N, Hitachi-High, Tokyo, Japan) [35]. The stomatal length (Lp), stomatal width (Wp), stomatal density (SD), stomatal size (SS), stomatal aperture (Sa) and maximum stomatal conductance (Gsmax) were calculated as follows:
Sa = π × Lp × Wp/4
(1)

SS = π × Lg × Wg/4-SA
(2)
(3)$$\mathrm{Gsmax}=d\;\times \;\alpha_{\max}\;\times \;\mathrm{SD}/\nu \;\times \;(\mathrm{Lp}+\pi /2\;\times \;\sqrt{\alpha_{\max}/\pi }$$

π = 3.14; d = 24.6 × 10^−6^ m^2^·s^−1^; α_max_: average maximum stomatal area; ν = 24.4 × 10^−3^ m^3^·mol^−1^.

#### 2.3.4. Photosynthetic Measurements

The contents of chlorophyll a (Chl a), chlorophyll b (Chl b), chlorophyll a+b (Chl a+b) and chlorophyll a/b (Chl a/b) in leaves were determined according to the method of Arnon et al. [34].

As described by Lin et al., the net photosynthetic rate (*Pn*), transpiration rate (*Tr*), intercellular CO_2_ concentration (*Ci*) and stomatal conductance (*Gs*) of the third leaf from top to bottom of each plant were quantified by photofuorometric apparatus (CIRAS-2, PP-system, Hitchin, UK). Each treatment was repeated three times [36].

As described by Yuan et al., the third fully developed leaf from the bottom of the M9-T337 plants was selected and subjected to a 30 min dark treatment. A green fluorescence imager (Imag-PAM, Heinz Walz GmbH, Bavaria, Germany) was used to measure the initial fluorescence (F0), maximum fluorescence (Fm), maximum photochemical quantum yield (Fv/Fm) and photochemical quenching coefficient (qP). Each treatment was repeated three times [37].

#### 2.3.5. Osmolytes

The relative conductivity (REC) was measured by conductivity meter [38]; malondialdehyde (MDA) was determined by thiobarbituric acid (TBA) method [39]; Pro was determined by acid ninhydrin staining method [40]; SS was determined by anthrone method [41]; SP was determined by Coomassie brilliant blue G-250 staining [42].

#### 2.3.6. Endogenous Hormone

As described by Zheng et al., the contents of IAA, GA_3_, ZT and ABA in 0.5 g leaves were determined by high performance liquid chromatography–tandem mass spectrometry (UHPLC–MS/MS). The chromatographic conditions were as follows: a Symmetry C18 chromatographic column (5 μm, 4.6 mm × 250 mm column), a mobile phase composed of methanol and 0.1% phosphoric acid at a volume ratio of 1:9, a flow rate of 1 mL/min, a detection wavelength of 254 nm, and a column temperature of 30 °C. The sample injection volume was 10 μL [43].

#### 2.3.7. Antioxidant Measurements and ASA-GSH Cycle

The activities of antioxidant enzymes (SOD, POD, and CAT), ASA-GSH cycle enzymes (APX, AAO, MDHAR, and GR) and the content of ASA-GSH cycle substances (ASA, GSH, and GSSG) were determined using the kits made by Beijing Soleibao Technology Co., Ltd. (Beijing, China). Each treatment was repeated 3 times.

#### 2.3.8. ROS

Superoxide anion free O_2_^•−^ was measured by the hydroxylamine oxidation method [44]. Fresh leaves (1 g) were added to 3 mL of phosphate grinding buffer (65 mmol·L^−1^), and the homogenate was centrifuged at 10,000× *g* for 15 min. The supernatant was collected in a 5 mL centrifuge tube to measure O_2_^•−^. H_2_O_2_ was determined by xylenol orange method [45]. Hydrogen peroxide and superoxide anion staining was determined as per Zhang et al. [46].

#### 2.3.9. Na^+^, K^+^, and Ca^2+^ Contents

The leaves of M9-T337 seedlings were dried and ground in an oven. Leaf powder samples (5 g) were accurately weighed and digested with H_2_SO_4_-H_2_O_2_ [47]. The contents of sodium and potassium ions were determined by flame photometer (FP6410, Perkin Elmer, Waltham, MA, USA).

#### 2.3.10. qRT‒PCR

Total RNA was extracted using an RNA extraction kit from Bi-oTeke, Beijing, China, and reverse transcription was performed using TaKaRa’s PrimeScriptTM RT kit (TaKaRa, Kyoto, Japan) and gDNA Eraser (Perfect Real Time). The primers were designed by Shanghai Sangong Bioengineering Co., Ltd. (Shanghai, China) and the sequences were obtained from the NCBI database (Table 1). The real-time quantitative PCR was performed using apple cDNA as a template and GAPDH as reference. The reaction system consisted of TB GreenTM Premix Ex Taq II (TaKaRa, Otsu, Japan), 10 μmol each of upstream and downstream primers, cDNA template and dd H_2_O in amounts of 10, 1, 1, 2 and 6 μL, respectively, and the reaction conditions included pre-denaturation at 95 °C for 3 min, followed by denaturation at 95 °C for 5 s, annealing at 56 °C for 30 s, and extension at 72 °C for 30 s. Forty cycles were repeated. Each sample was triple-treated to ensure the reliability of the results. The data obtained from quantification were analyzed by 2^−∆∆CT^ method.

#### 2.3.11. Statistical Analysis

Microsoft Office Excel 2019 was used for data processing. SPSS 22.0 (IBM, Armonk, NY, USA) and Origin 9.1 (OriginLab, Hampton, MA, USA) software were used for significance tests, correlation and principal component analysis. Duncan method was used to test the difference in significance level (*p* = 0.05).

## 3. Results

### 3.1. Characterization of ZnO NPs

To ascertain whether the ZnO NPs used in this experiment meet the requirements for nanomaterials, the morphology and characterization of ZnO NPs were observed. TEM observation results were found to show that the ZnO NPs particles are approximately elliptical or spherical in shape (Figure 1a), distributed mainly in aggregated forms, and that the average particle size was 20–30 nm (Figure 1b).

### 3.2. Effects of ZnO NPs on Plant Phenotype and Root Phenotype of M9-T337 Under Saline Alkali Stress

As shown in Figure 2, The leaves of M9-T337 seedlings treated solely with the SA treatment mostly exhibited significant chlorisis and wilting, with overall dwarfism and severely hindered root development. In contrast, the growth conditions of M9-T337 seedlings pretreated with ZnO NPs at different concentrations (T1~T4) were significantly better than those under the SA treatment. Among these, the T3 treatment showed the best leaf regreening effect, the best growth condition, and the most vigorous root development, followed by T2, T4, and T1 in descending order. Under the SA treatment, the total root length, average diameter, volume, surface area, number of tips, and root activity were significantly lower (Table 2). After ZnO NP treatment, the above index values have increased, and the effect of the T3 treatment has become the most obvious. Compared with the SA treatment, the total root length increased by 97.30%, the diameter increased by 62.50%, the volume increased by 176.23%, the surface area increased by 92.88%, the number of tips increased by 117.33%, and the root activity increased by 10.28%.

### 3.3. The Effects of ZnO NPs on the Growth Parameters of M9-T337 Leaf Seedlings Under Saline Alkali Stress

As shown in Table 3, under the SA treatment, the fresh and dry weights of the whole plant, aboveground part, and roots of M9-T337 seedlings were significantly lower than those of the CK treatment. However, these parameters tended to increase after pretreatment of ZnO NPs (T1~T4). Specifically, the maximum fresh (dry) weights of the whole plant and roots were achieved under the T3 treatment, with increases of 117.83% and 190.94% in fresh weight, and 97.96% and 147.95% in dry weight, respectively, compared with the SA treatment. The maximum fresh (dry) weights of the aboveground part were obtained under the T2 treatment, with increases of 66.45% and 1.09%, respectively, compared with the SA treatment. As illustrated in Figure 3A–D, saline alkali stress significantly inhibited the growth rate of M9-T337 seedlings, with plant height, stem diameter, total leaf area, and leaf perimeter all being significantly lower than those of the CK treatment. However, after the application of ZnO NPs, these parameters were significantly increased compared with the SA treatment, and the parameters were the best under the T3 treatment. Specifically, compared with the SA treatment, plant height increased by 1.38 times, stem diameter increased by 1.24 times, total leaf area increased by 1.30 times, and leaf perimeter increased by 1.17 times.

### 3.4. Effects of ZnO NPs on Stomatal Morphology of M9-T337 Seedlings Under Saline Alkali Stress

Stomata serve as crucial gateways for carbon and water exchange between plant leaves and the atmosphere. As shown in Figure 4, under the SA treatment, most stomata are in a closed state. However, the closure of pores was found to be alleviated after ZnO NP treatment. The stomatal opening state was the under the T3 treatment (150 mg/L ZnO NPs).

The stomatal structural parameters of M9-T337 seedling leaves under various treatments were calculated and are presented in Table 4. Under the SA treatment, the Lp, Wp, SD, SS, Sa, and Gsmax of M9-T337 seedlings were the lowest. The indicators of different concentrations of ZnO NPs pretreatment showed a significant concentration effect. Specifically, the values of Lp, Wp, SD, SS and Gsmax under the T3 treatment reached their respective peaks of 14.57 μm, 1.82 μm, 266.00 mm^2^, 16.51 μm^2^, and 0.96, which were 56.16%, 188.89%, 155.09%, 185.64% and 123.26% higher than those under the SA treatment. The Sa reached its maximum under the T2 treatment, being 1.35 times that of the SA treatment.

### 3.5. Effects of ZnO NPs on the Photosynthesis of M9-T337 Seedlings Under Saline Alkali Stress

Photosynthetic pigments (Chl a, Chl b, and Chl a+b) serve as indicators of photosynthetic intensity in plants. As illustrated in Figure 5A–D, compared with the CK treatment, the SA treatment resulted in reductions of 31%, 39%, 33%, and 29% in Chl a, Chl b, Chl a+b contents, and Chl a/b, respectively. However, following pretreatment with ZnO NPs (T1 to T4), the contents of these pigments exhibited a trend of initial increase followed by decrease. Specifically, under the T3 treatment, Chl a, Chl b, Chl a+b, and Chl a/b reached peak values of 14.90 mg·g^−1^, 27.30 mg·g^−1^, 42.20 mg·g^−1^, and 0.55, respectively, representing decreases of only 35.43%, 19.27%, 28.41%, and 17.51% compared with the CK treatment. These results demonstrate the potential of ZnO NPs, especially when the concentration was 150 mg/L, to enhance the accumulation of photosynthetic pigments to in turn improve the photosynthetic capacity of M9-T337 leaves under saline alkali stress.

Furthermore, under the SA treatment, the *Pn*, *Gs*, and *Tr* of M9-T337 leaves were significantly lower than those of the CK treatment, being 0.44, 0.34, and 0.47 times that of the CK treatment, respectively. Meanwhile, the *Ci* was significantly higher than that of the CK treatment, being 1.73 times that of the CK treatment. As shown in Figure 6A–C, after ZnO NPs pretreatment, *Pn*, *Gs*, and *Tr* all increased significantly, while *Ci* exhibited a decreasing trend (Figure 6D). Specifically, under the T3 treatment, *Pn*, *Gs*, and *Tr* were significantly higher than those under other treatments, with decreases of 25.68%, 32.52%, and 20.01% compared with the CK treatment, respectively. In contrast, *Ci* under the T3 treatment was the lowest, with a decrease of only 15.14% compared with the SA treatment alone.

Finally, the effects of saline alkali stress on chlorophyll fluorescence parameters (Fm, F0, Fv/Fm, and qP) in M9-T337 leaves were investigated. As shown in Figure 7A–D, under the SA treatment, the values of Fm, F0, Fv/Fm, and qP were significantly reduced, being 0.60, 0.39, 0.76, and 0.45 times those of the CK treatment, respectively. However, after ZnO NP pretreatment, the values of Fm, F0, Fv/Fm, and qP in M9-T337 leaves increased to varying degrees, increasing first and then decreasing with the concentration of ZnO NPs. Among these, the F0, Fm, Fv/Fm and qP values under the T3 treatment were the largest, being 0.42,0.19,0.61 and 23.25, respectively, showing respective decreases of 20.85%, 48.36%, 12.07% and 45.82% when compared with the CK treatment.

### 3.6. Effects of ZnO NPs on Plant Growth and Membrane Lipid Oxidation of M9-T337 Seedlings Under Saline Alkali Stress

As illustrated in Figure 8A, the RWC of leaves decreased by 33.77% under the SA treatment, while the RWC increased after pretreatment with different concentrations of ZnO NPs (T1–T4), being 0.63, 0.68, 0.73 and 0.65 times that of the CK treatment, respectively. Additionally, the relative water content under the T3 treatment was highest. According to Figure 8B, the water use efficiency of M9-T337 seedling leaves under saline alkali stress treatment was significantly lower than that of the CK treatment, which was 0.58 times that of the CK treatment. However, the water use efficiency was significantly improved after pretreatment with ZnO NPs, and there was no significant difference compared with the CK treatment, and both remained at about 2%.

As shown in Figure 9A–D, saline alkali stress was found to significant enhance the production of O_2_^•−^ and H_2_O_2_, resulting in an increase of 143.12% in MDA content and an increase of 158.90 in REC. The pretreatment of ZnO NPs (T1~T4) reduced the H_2_O_2_ content, O_2_^•−^ production rate, REC content and MDA content of M9-T337 seedling leaves, which all reached their lowest under the T3 treatment, being 35.84%, 21.20%, 20.13% and 30.10% lower than those of the SA treatment, respectively. After 45 days of stress, the leaves of M9-T337 seedlings under each treatment were stained with NBT and DBA (Figure 10A,B). The staining results were found to show that the accumulation of NBT and DBA staining under the T3 treatment was the least, which is consistent with the determination results of the O_2_^•−^ production rate and H_2_O_2_ content in M9-T337 seedling leaves.

In addition, under the SA treatment, the enzyme activities of SOD, POD, and CAT in the leaves of M9-T337 seedlings were significantly increased (Figure 9E–G), being 1.22 times, 1.29 times and 1.93 times that of the CK treatment, respectively. Additionally, the application of ZnO NPs significantly enhanced the activities of SOD, POD, and CAT in the leaves of M9-T337 seedlings. Specifically, the SOD activities pretreated with different concentrations of ZnO NPs (T1–T4) increased by 21.60%, 36.30%, 44.67%, 53.51% and 41.34% compared with CK, respectively, and the POD activities were 1.29, 1.52, 1.65, 1.75 and 1.51 times that of CK, respectively. CAT activity increased by 23.33%, 28.23%, 36.86% and 22.24% compared with the SA treatment, respectively. In general, the activities of SOD, POD and CAT were strongest under the T3 treatment.

### 3.7. Effects of ZnO NPs on Glutathione Cycle-Related Substances and Enzyme Activities in Leaves of M9-T337 Seedlings Under Saline Alkali Stress

ASA, GSH, and GSSG are substances related to the ASA-GAH cycle. As shown in Figure 11A–C, under the SA treatment alone, ASA content decreased significantly, being 0.27 times that of the CK treatment, while GSH and GSSG content increased significantly, which was 2.60 and 1.53 times that of the CK treatment. After pretreatment with different concentrations of ZnO NPs, the ASA content of each treatment (T1~T4) increased first and then decreased. The ASA content of the T3 treatment increased to the highest, 32,061.54 nmol·g^−1^, which was 131.56% higher than that of the SA treatment. In contrast, the GSH and GSSG contents showed a trend of first decreasing and then increasing, both reaching their lowest levels under the T3 treatment at 479.31 nmol·g^−1^ and 998.15 nmol·g^−1^, respectively, being 75.91% and 38.51% lower than those under the SA treatment.

In addition, the effect of ASA-GAH cycle enzyme activity was studied by measuring the activity of AAO, APX, GR, and MDHAR (Figure 11D–G). Under the SA treatment, AAO and MDHAR activities were significantly reduced, being 0.76 and 0.78 times that of the CK treatment, respectively. Additionally, APX and GR activities were significantly increased, being 3.96 and 2.38 times that of the CK treatment, respectively. After pretreatment with ZnO NPs, the activities of AAO, MDHAR, and GR increased to their highest under the T3 treatment, with those indexes increasing by 103.03%, 123.48% and 151.88%, respectively, compared with the SA treatment. However, the activity of APX was the highest under the SA treatment. After ZnO NP treatment, the activity of APX treated with T3 only decreased by 11.93%. These findings indicate that exogenous ZnO NPs play an important role in enhancing the activities of enzymes related to the ASA-GSH cycle in M9-T337 seedlings under saline alkali stress.

### 3.8. The Effect of ZnO NPs on the Content of Osmotic Adjustment Substances in Leaves of M9-T337 Seedlings Under Saline Alkali Stress

As shown in Figure 12A, SS content in the leaves of M9-T337 seedlings was lowest under the SA treatment, being 0.55 times that of the CK treatment. After pretreatment with ZnO NPs, SS increased to varying degrees. Specifically, SS increased first and then decreased among the T1–T4 treatments, with values of 13.01,14.30,16.80 and 10.92 mg·g^−1^, respectively, being 37.41%, 31.20%, 19.14% and 47.45% lower than the CK, respectively. The T3 treatment had the lowest decrease.

As illustrated in Figure 12B–D, under the SA treatment, the contents of SP, Pro, and St in the leaves of M9-T337 seedlings are significantly higher than those of the CK treatment by respective orders of 1.90, 1.41 and 1.35 times. The contents of SP, Pro and St increased after ZnO NPs pretreatment. Among these, the contents of SP, Pro and St were the highest under the T3 treatment, being 31.10, 30.78 and 7.98 mg·g^−1^, respectively, amounting to 2.84, 2.18 and 2.06 times those of the CK treatment.

### 3.9. Effects of ZnO NPs on Endogenous Hormone Content in Leaves of M9-T337 Seedlings Under Saline Alkali Stress

The effect of ZnO NPs on the endogenous hormone content in the leaves of M9-T337 seedlings is shown in Figure 13A–D. Under the SA treatment, ZT, IAA, and GA_3_ in the leaves of M9-T337 seedlings were significantly reduced, being 0.31, 0.45, and 0.60 times those of the CK treatment, respectively. After pretreatment with ZnO NPs, the contents of ZT, IAA and GA_3_ reached their peaks under the T3 treatment. Specifically, compared with the SA treatment, the contents of ZT, GA_3_, and IAA under the T3 treatment increased by 189.05%, 114.41%, and 48.14%, respectively. Conversely, the content of ABA reached its highest level under the SA treatment, being 3.02 times that of the CK treatment. After the pretreatment of ZnO NPs, the ABA content under the T3 treatment was the lowest, at 246.55 ng·g^−1^, which was 0.56 times that of the SA treatment.

### 3.10. The Effects of ZnO NPs on the Contents of Sodium, Potassium and Calcium Ions in Leaves of M9-T337 Seedlings Under Saline Alkali Stress

As shown in Figure 14A–D, under the SA treatment, the contents of Na^+^ and the Na^+^/K^+^ ratio were significantly higher than those of the CK treatment. However, pretreatment with ZnO NPs reduced the contents of Na^+^ and of the Na^+^/K^+^ ratio. Specifically, both the Na^+^ content and the Na^+^/K^+^ ratio reached their lowest levels under the T3 treatment, with values of 0.8 g·kg^−1^ and 0.18, respectively, accounting for only 36.36% and 8.62% of SA. Additionally, under the SA treatment, the contents of Ca^2+^ and K^+^ were significantly lower than those in the CK treatment. Pretreatment with ZnO NPs at different concentrations (T1 to T4) exhibited a trend of first increasing and then decreasing for both Ca^2+^ and K^+^ contents in the leaves of M9-T337 seedlings. Specifically, the contents of Ca^2+^ (16.65 mg·g^−1^) and K^+^ (4.43 g·kg^−1^) in leaves under the T3 treatment were the highest, being 1.51 and 2.16 times that of the SA treatment, respectively.

### 3.11. Effects of Nano-Zinc Oxide on the Expression of Saline Alkaline Response Genes in M9-T337 Seedlings

As shown in Figure 15, the expression levels of Na^+^ transporter genes (*MdCAX5*, *MdSOS*, *MdALT1* and *MdCHX15*) and K^+^ transporter gene (*MdSKOR*) in leaves were significantly down-regulated under the SA treatment, while the expression levels of K^+^ transporter gene (*MdNHX4*) and three antioxidant enzymes (*MdPOD*, *MdCAT* and *MdSOD*) were significantly up-regulated. However, after pretreatment with ZnO NPs, the expression levels of Na^+^ transport genes did not change significantly compared with the CK treatment, the expression levels of three antioxidant enzyme regulatory genes and *MdNHX4* decreased, and the expression level of *MdSKOR* increased.

### 3.12. Comprehensive Evaluation of the Effects of Different Concentrations of ZnO NPs on the Growth and Physiological Effects of M9-T337 Seedlings Under Saline Alkali Stress

#### Correlation, Principal Component Analysis (PCA) and Composite Score Ranking

In order to comprehensively judge the effects of different concentrations of ZnO NPs on the growth and physiological indexes of M9-T337 apple rootstock seedlings under the SA treatment, SPSS statistical analysis software was used to perform correlation analysis and principal component analysis on 37 symbolic indexes such as *Pn*, *Ci* and *Tr* (Figure 16A). The results were found to show that the *Pn* of M9-T337 seedling leaves was significantly positively correlated with *Tr*, *Gs*, F0, Chl a, Chl b, Chl a+b, Chl a/b, RWC, ZT and IAA (*p* < 0.01), significantly negatively correlated with *Ci*, MDA, REC, H_2_O_2_, O_2_^•−^ and ABA (*p* < 0.01), significantly positively correlated with Fm, Fv/Fm, qP, ASA and GA_3_ (*p* < 0.05), and significantly negatively correlated with SS (*p* < 0.05).

In addition, the evaluation index values of each index are standardized, the data of the negative correlation index are processed by 1-X, and the three principal components with the characteristic value of the level variable >1 are screened. The maximum characteristic values are 24.617,10.128 and 1.3357, and the variance contribution rates are 66.531%, 27.372% and 3.667%, respectively. Additionally, the cumulative variance contribution rate reaches 97.570%, which meets the analysis requirements (Figure 16B). The comprehensive ranking (F) is the sum of the product of each principal component score and the current corresponding contribution rate, as follows: F = F1 × 66.531% + F2 × 27.372% + F3 × 3.667%. The scores of each treatment are 1.0153 (CK), −0.9554 (SA), −0.5258 (T1), −0.0717 (T2), 0.6041 (T3), −0.0665 (T4). Therefore, the comprehensive ranking of the effects of different concentrations of ZnO NPs on the growth and physiological characteristics of M9-T337 seedlings under saline alkali stress was found to be CK > T3 > T4 > T2 > T1 > SA. Comprehensive analysis showed that the T3 treatment had the best effect.

Finally, cluster analysis of M9-T337 seedlings under ZnO NP treatment was carried out based on 37 growth and physiological parameters using Euclidean distance and the inter-group connection clustering method. The results (Figure 16C), at the Euclidean distance of 0.8, can be divided into three categories. Cluster I includes Chl a, Chl a+b, *Pn*, qP, *Gs*, Fv/Fm, F0, IAA, and Chl a/b. Cluster II includes *Ci*, REC, H_2_O_2_, GSH and APX. Cluster III includes Pro, POD, St, water utilization and MDHAR. Comprehensive analysis showed that Chl a, *Ci*, APX, Pro, and MDHAR could be used as representative indexes in this experiment, which could provide some reference for the screening of physiological indexes of M9-T337 seedlings by different ZnO NPs in the future.

## 4. Discussion

Roots are instrumental in enabling plants to acclimate to diverse environmental conditions by modulating the uptake and translocation of water and nutrients. When plants encounter saline alkali stress, their root systems initiate stress signaling cascades. These signals are transmitted to the aerial parts of the plant, subsequently affecting their normal growth and development, such as by causing plant wilting, leaf chlorisis, and pigment degradation [56]. Chlorophyll, a pivotal constituent of photosynthetic pigments, functions to harness and transform external light energy to sustain the photosynthetic activity of plant leaves [57]. Studies have shown that, under saline stress, chlorophyll synthesis in plants is blocked and the rate of degradation increases, ultimately leading to a decrease in chlorophyll content [45]. The results of this experiment show that the accumulation of biomass, Chl a, Chl b, Chl a/b and Chl a+b are significantly higher in plants treated with ZnO NPs under saline alkali stress than in plants treated with saline alkali stress alone [58]. This shows that ZnO NPs can stabilize the soil osmotic pressure and maintain the intracellular ion balance, thereby enhancing the capture and utilization of light energy by plants, promoting the synthesis of chlorophyll, and ultimately leading to the enhancement of photosynthesis and growth and development of M9-T337 seedlings. At the same time, Kasivelu et al. have also found that ZnO NPs can up-regulate the expression of plant photosystem-related genes, promote plant photosynthetic metabolism and chlorophyll synthesis, increase biomass accumulation, stimulate root development, and ultimately reduce the damage of saline alkali stress to plants [59]. However, in the study of He et al., the application of ZnO NPs did not significantly increase the chlorophyll content, and even showed a downward trend, which may be related to the dosage and concentration of ZnO NPs [60].

Photosynthetic parameters are the basis of plant survival and growth. Under saline alkali stress, the excessive accumulation of Na^+^ will disturb the normal regulation mechanism of stomata, thus limiting the photosynthesis of plants [61]. Dang et al. have shown that, under stress conditions, photosynthesis is limited in two major ways: stomatal limitation, which is due to a decrease in stomatal conductance and results in an insufficient concentration of intercellular CO_2_ to meet the demands of photosynthesis, and non-stomatal limitation, which is mainly due to a decrease in chloroplast activity, the decrease in the activity of ribulose-1,5-bisphosphate carboxylase (Rubisco) and the reduced regeneration of ribulose-1,5-bisphosphate (RuBP) [31]. Ayari et al. have also shown that, if *Pn*, *Gs* and Tr had the same trend as *Ci*, the photosynthesis of plants is limited by stomata under saline alkali stress, otherwise it is limited by non-stomatal factors [62]. In this experiment, with the increase of ZnO NP concentration, the change trend of *Pn*, *Gs* and *Tr* was found to be opposite to that of *Ci*, indicating that the photosynthesis of M9-T337 seedlings is limited by non-stomatal factors, which is consistent with the results of Dhokne et al. [63]. This phenomenon may be related to the damage to the thylakoid membrane caused by salt ion accumulation, which affects the synthesis and decomposition of chlorophyll and the rate of photosynthetic electron transport. In addition, the study of Linglan et al. in cowpea further confirmed that ZnO NPs can effectively alleviate the stomatal limitation of plant photosynthesis by increasing the opening and closing degree of stomata, enhancing the carbon fixation efficiency of cowpea and reducing the concentration of intercellular CO_2_, which is consistent with the results of this experiment [64,65,66].

In the complex mechanism of plant photosynthesis, PSII is the photosynthetic structure most affected by photoinhibition, and its excessive consumption can lead to functional damage of plant photosynthetic organs [67]. Among these, chlorophyll fluorescence parameters, as a constant and as a variable by which to accurately quantify the state of plant photosynthesis, can reflect the absorption, transfer and transmission mechanism of electrons in the process of photosynthesis, and have important reference value for further understanding the mechanism of plant photosynthesis [68]. In this study, saline alkali stress reduced the F0, Fm, Fv/Fm and qP of M9-T337 seedlings, which is consistent with the results of Mehta et al. [69]. This may be because saline alkali stress can damage the structural integrity and functional efficiency of PSII, reduce the operation efficiency of the electron transport chain, and then weaken the photochemical activity of PSII. In addition, these parameters were significantly improved after adding ZnO NPs, which is consistent with the results of Yan et al. This effect may be attributed to the fact that Zn, as a key element in activating various enzyme activities in photosynthesis in its ionic form Zn^2+^, can further repair the damaged PSII structure by promoting the activity of peptidase and protein in the matrix [70,71]. Meanwhile, Chen et al. found that ZnO NPs can alleviate the damage caused by salt stress on the PS II photosystem of tea tree by balancing the electron transfer from the donor side, acceptor side, and reaction centers of PS II photosystem in their study [72]. Liu et al. have also demonstrated in spinach that the application of ZnO NPs significantly up-regulates the expression of light-harvesting complex II (LHCB)-related genes [73].

Saline alkali stress disrupts the balance between photosynthetic electron transport and the Calvin cycle, causing electrons to transfer from chloroplasts and mitochondria to oxygen molecules, leading to increased ROS, particularly H_2_O_2_ and O_2_^•−^ [74]. High ROS levels increase cell membrane permeability and accelerate lipid peroxidation, ultimately damaging or killing plants [75]. MDA and REC are key indicators of cell damage, their increased levels compromise membrane integrity and hinder plant growth [74]. In this experiment, MDA and REC levels rose under saline alkali stress but decreased after ZnO NP treatment, consistent with Liang et al. ’s findings [76]. This suggests that ZnO NPs enhance plants’ ability to manage ion balance by regulating ion channels and transporters, thereby reducing salt stress effects. Weisany et al. also found that ZnO NPs can regulate ROS scavenging genes, inhibiting lipid peroxidation and alleviating stress damage to plants [77].

Plants can eliminate ROS by increasing the activity of antioxidant enzymes, thereby balancing ROS production and removal [78]. The primary antioxidant enzymes in plants are SOD, POD, and CAT. SOD, the first line of defense for maintaining cell membrane integrity, decomposes excessive ROS into H_2_O_2_ and O_2_^•−^. Then, POD and CAT work together to convert H_2_O_2_ and O_2_^•−^ into oxygen and water, which are excreted, thus restoring the enzyme system’s balance [79]. The results of this experiment show that the activities of SOD and POD in leaves increased first and then decreased with the prolongation of saline alkali stress time, which is consistent with the study of Malus halliana by Rrary [80]. This indicates that plants could adjust the activity of antioxidant enzymes in the body to maintain the balance of reactive oxygen species in the short term of stress, but with the aggravation of stress, the enzyme activity decreased and the resistance gradually decreased. After pretreatment with ZnO NPs, the activity of SOD and POD increased, which is consistent with the results of Alabdallah et al. [81]. This may be because the ZnO NPs activates the antioxidant enzyme system of plants and enhances the expression and activity of these key enzymes to remove harmful reactive oxygen species, thereby improving the resistance of plants to oxidative stress.

The ASA-GSH cycle is an important part of the plant antioxidant system. Environmental stress and exogenous substances can affect the adaptability of plants to stress by affecting the metabolic level of the ASA-GSH cycle [82]. Studies have shown that the ASA-GAH cycle can use the catalysis of APX and the reduction of NADPH in GSH to react with H_2_O_2_ to remove reactive oxygen species free [83]. This study has shown that the activities of APX and MDHAR in the leaves of M9-T337 apple seedlings decrease under saline alkali stress while the activities of AAO and GR increase, which is consistent with the results of Wu et al. in maize [84]. Plants can adapt to saline alkali environment by increasing the content of ASA-GSH-related substances and enzyme activity. After treatment with ZnO NPs, the contents of APX, GR and MDHAR increased, which is consistent with the study of tan et al., indicating that ZnO NPs could provide sufficient substrates for the ASA-GSH cycle and improve the ability to eliminate reactive oxygen species, thus reducing the degree of oxidative damage to plants [85]. Meanwhile Prakash et al. have shown, in rice, that the rise in ASA, GSH content and related enzyme activities could be attributed to ZnO NPs facilitating the conversion of ASA to DHA and the equilibrium exchange between GSH and GSSG, which maintains the redox homeostasis of the leaves and enhances their salinity tolerance [86].

Soluble sugar, soluble protein and proline are the key osmotic adjustment substances in the process of stress resistance. Among them, soluble protein, due to its strong water holding capacity, promotes the stability of the cell membrane to a certain extent. Proline and soluble sugar, as osmotic regulators, can stabilize the cell membrane and scavenge reactive oxygen species by increasing its content under adversity. They can also be used as intracellular cascade messengers to improve tolerance to various stresses [87]. The results of this experiment show that the SP content of M9-T337 seedlings demonstrate a downward trend under saline alkali stress, while the Pro content showed a gradual increase trend. The contents of SP, SS and Pro in the leaves of seedlings pretreated with ZnO NPs were significantly higher than those under saline alkali stress, which is consistent with the research of Soliman et al. regarding maize [88]. This may be due to the fact that ZnO NPs regulate the activity of proline synthase, glycolytic enzyme and glycosidase, thus providing antioxidant protection. In addition, in the study of Zhang et al., ZnO NPs were found to be able to enhance the stress resistance of plants through a variety of applications, and are thus better able to maintain normal physiological activities under saline alkali stress conditions, such as by reducing protein degradation, enhancing plant osmotic adjustment ability, and promoting protein synthesis in plants [89].

Endogenous hormones, crucial signaling molecules in both biotic and abiotic stress responses, significantly influence plant growth and development. They regulate ROS production and clearance by affecting intracellular pathways like the MAPK pathway [90]. In this study, under saline alkali stress, the levels of ZT, GA_3_, and IAA in M9-T337 seedling leaves decreased, while ABA levels increased, aligning with findings of Xu et al. regarding wheat [91]. However, after ZnO NP pretreatment, ZT, GA_3_, and IAA levels were found to rise, and ABA levels fall, consistent with the results of Jia et al. [92]. This suggests that ZnO NPs may enhance plant tolerance to saline alkali stress by regulating hormone levels, affecting growth, development, cell division, and stress response, though the exact mechanisms require further investigation. IAA metabolism is vital for plant growth. Wang et al. found that stress environments inhibit Try-to-IAA conversion, reducing free IAA and hindering growth, matching this study’s results [93]. ZnO NP application increased IAA levels, indicating the potential promotion of Try-to-IAA conversion enzymes, as seen in Chen et al.’s jujube tree study [82].

Ion homeostasis is one of the mechanisms of salt tolerance in plants. Usually, the K^+^/Na^+^ ratio in the cytoplasm is maintained at about 1, but can become lower than 1 when under stress. Therefore, maintaining a high K^+^/Na^+^ ratio is an important mechanism by which to maintain the normal growth of most plants [94]. Hasegawa et al. studies have shown that, under saline alkali stress, the excessive absorption of Na^+^ by plants will significantly inhibit the absorption of K^+^ and Ca^2+^. This antagonistic effect can destroy the normal function of cell membrane, resulting in insufficient nutrients in plants [95]. The results of this experiment show that the content of sodium ions increased gradually and the contents of K^+^ and Ca^2+^ decreased gradually with the prolongation of saline alkali stress time. After ZnO NP treatment, the increase rate of Na^+^ decreased, while the contents of K^+^ and Ca^2+^ increased, which is consistent with the results of Anschütz et al. [81]. This may be because ZnO NPs facilitate K^+^ and Ca^2+^ transport and distribution in plants while limiting Na+ long-distance transport, leading to increased K^+^ and Ca^2+^ and decreased Na^+^ levels. Alternatively, ZnO NPs can enhance plant root ion-selective absorption, boosting K^+^ uptake and reducing Na^+^ absorption, thereby maintaining potassium–sodium balance.

## 5. Conclusions

ZnO NPs (particularly at a concentration of 150 mg/L) significantly enhanced the saline alkaline tolerance of M9-T337 seedlings, which was manifested in several aspects, including their participation in osmotic regulation, oxidative adjustment, photosynthetic rate, ion homeostasis and hormone regulation. Although studies have confirmed that ZnO NPs have a positive effect on enhancing the salt tolerance of M9-T337 seedlings, the mechanism of ZnO NPs with ZT, IAA, GA_3_, and ABA still needs further study. In summary, understanding the intervention pathways and potential physiological and biochemical reactions of ZnO NPs plays a crucial role in improving the tolerance of ZnO NPs to saline alkali stress in M9-T337 seedlings.

## Figures and Tables

**Figure 1 plants-14-00233-f001:**
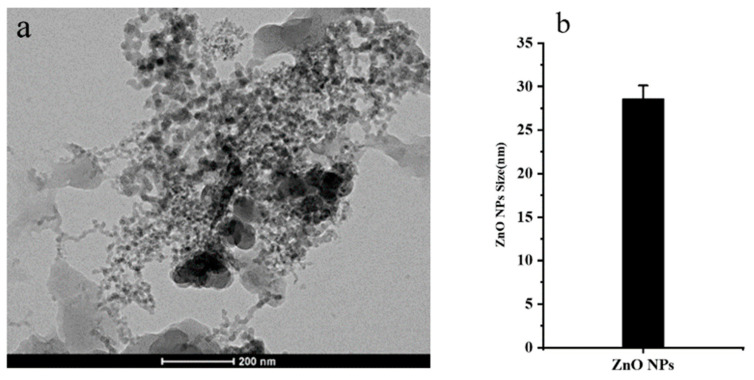
Transmission electron microscopy (TEM) imaging and TEM grain size measurements of ZnO NPs. (**a**) Transmission electron microscopy (TEM) imaging of ZnO NPs; (**b**) TEM particle size of ZnO NPs.

**Figure 2 plants-14-00233-f002:**
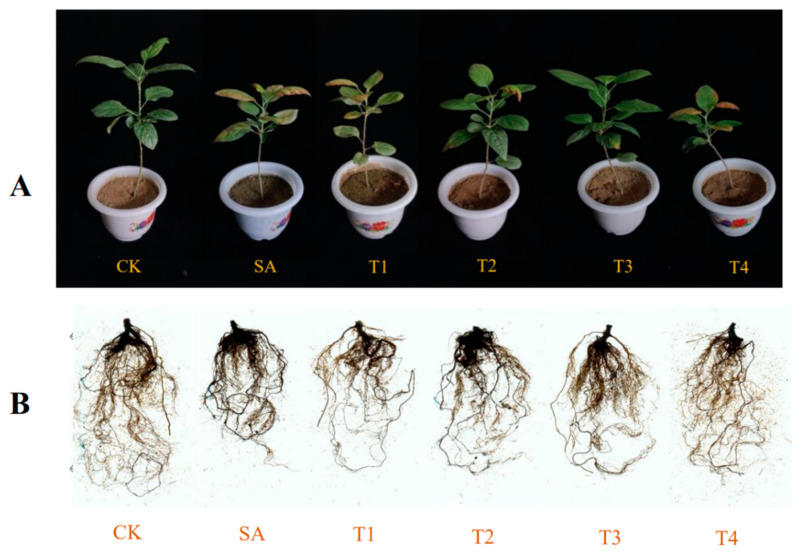
Effect of zinc oxide nanoparticles on leaves and roots of M9-T337 seedlings under saline stress. CK: normal treatment (control); SA: 100 mmol/L NaCl + NaHCO_3_ saline alkali stress; T1: saline alkali stress +50 mg/L ZnO NPs; T2: saline alkali stress +100 mg/L ZnO NPs; T3: saline alkali stress +150 mg/L ZnO NPs; T4: saline alkali stress +200 mg/L ZnO NPs. (**A**) Plant phenotype; (**B**) root phenotype.

**Figure 3 plants-14-00233-f003:**
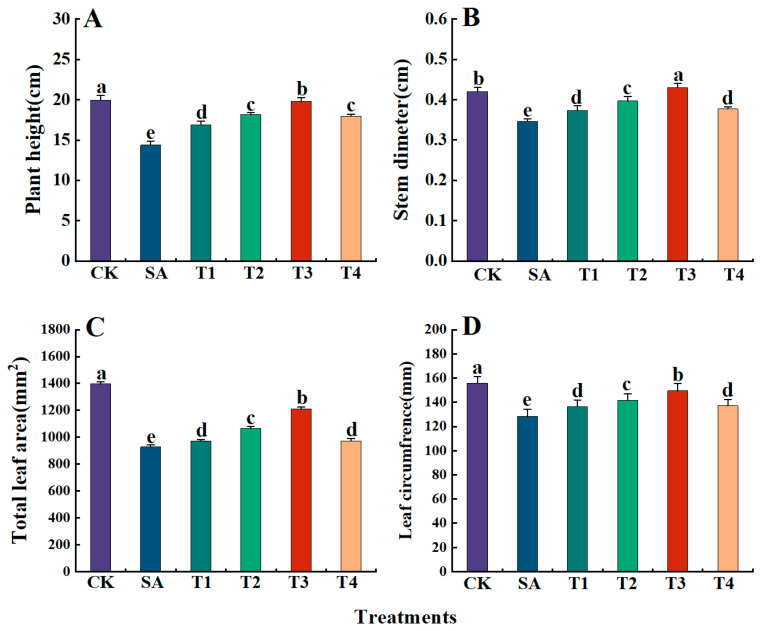
Effects of ZnO NPs on the growth parameters of M9-T337 seedlings under saline alkali stress. CK: normal treatment (control); SA: 100 mmol/L NaCl + NaHCO_3_ saline alkali stress; T1: saline alkali stress +50 mg/L ZnO NPs; T2: saline alkali stress +100 mg/L ZnO NPs; T3: saline alkali stress +150 mg/L ZnO NPs; T4: saline alkali stress +200 mg/L ZnO NPs. (**A**) plant height; (**B**) stem thickness; (**C**) leaf area; (**D**) leaf circumference. Vertical bars represent the standard errors of the means of three replicates. Data show the mean ± SE (n = 3). Different lowercase letters indicate significant differences between treatments with *p* < 0.05.

**Figure 4 plants-14-00233-f004:**
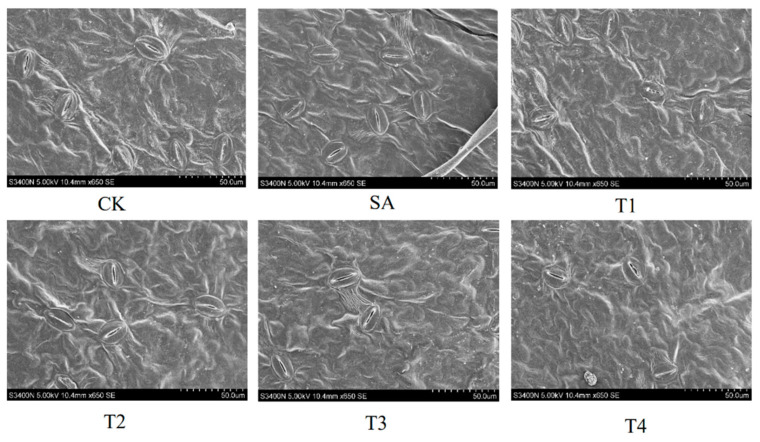
Effect of ZnO NPs on stomatal morphology and structure of leaves of M9-T337 seedlings under saline alkali stress. CK: normal treatment (control); SA: 100 mmol/L NaCl + NaHCO_3_ saline alkali stress T1: saline alkali stress +50 mg/L ZnO NPs; T2: saline alkali stress +100 mg/L ZnO NPs; T3: saline alkali stress +150 mg/L ZnO NPs; T4: saline alkali stress +200 mg/L ZnO NPs.

**Figure 5 plants-14-00233-f005:**
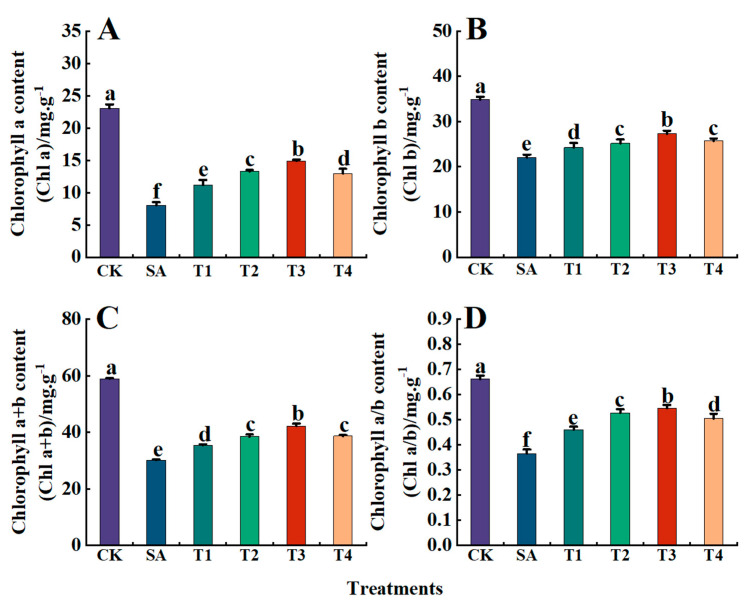
Effect of ZnO NPs on photosynthetic pigment content of M9-T337 seedling leaves under saline stress. CK: normal treatment (control); SA: 100 mmol/L NaCl + NaHCO_3_ saline alkali stress; T1: saline alkali stress +50 mg/L ZnO NPs; T2: saline alkali stress +100 mg/L ZnO NPs; T3: saline alkali stress +150 mg/L ZnO NPs; T4: saline alkali stress +200 mg/L ZnO NPs. (**A**) Chl a content; (**B**) Chl b content; (**C**) Chl a+b content; (**D**) Chl a/b content. Vertical bars represent the standard errors of the means of three replicates. Data show the mean ± SE (n = 3). Different lowercase letters indicate significant differences between treatments with *p* < 0.05.

**Figure 6 plants-14-00233-f006:**
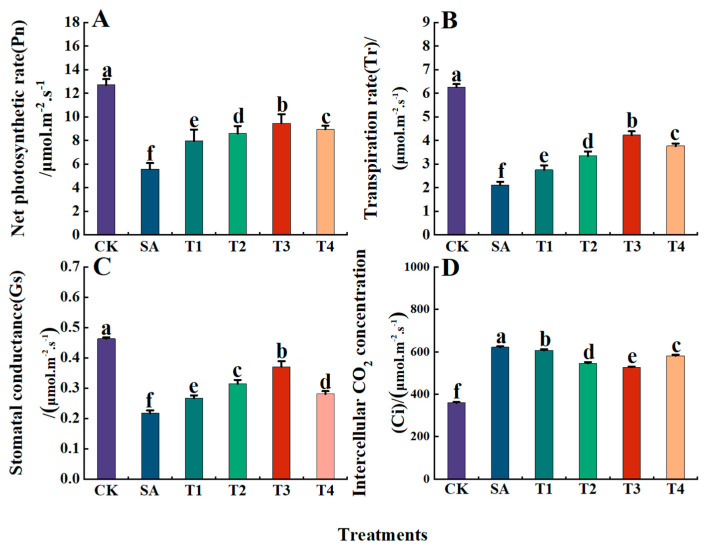
Effects of ZnO NPs on the photosynthetic characteristics of M9-T337 seedling leaves under saline stress. CK: normal treatment (control); SA: 100 mmol/L NaCl + NaHCO_3_ saline alkali stress; T1: saline alkali stress +50 mg/L ZnO NPs; T2: saline alkali stress +100 mg/L ZnO NPs; T3: saline alkali stress +150 mg/L ZnO NPs; T4: saline alkali stress +200 mg/L ZnO NPs. (**A**) *Pn*; (**B**) *Tr*; (**C**) *Gs*; (**D**) *Ci*. Vertical bars represent the standard errors of the means three replicates. Data show the mean ± SE (n = 3). Different lowercase letters indicate significant differences between treatments with *p* < 0.05.

**Figure 7 plants-14-00233-f007:**
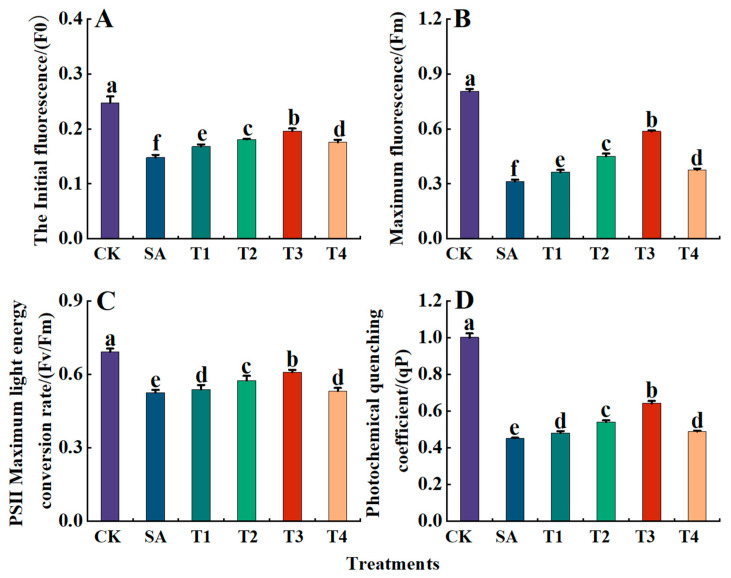
Effect of ZnO NPs on fluorescence parameters of M9-T337 seedling leaves under saline stress. CK: Normal treatment (control); SA: 100 mmol/L NaCl + NaHCO_3_ saline alkali stress; T1: saline alkali stress +50 mg/L ZnO NPs; T2: saline alkali stress +100 mg/L ZnO NPs; T3: saline alkali stress +150 mg/L ZnO NPs; T4: saline alkali stress +200 mg/L ZnO NPs. (**A**) F0; (**B**) Fm; (**C**) Fv/Fm; (**D**) qP. Vertical bars represent the standard errors of the means three replicates. Data show the mean ± SE (n = 3). Different lowercase letters indicate significant differences between treatments with *p* < 0.05.

**Figure 8 plants-14-00233-f008:**
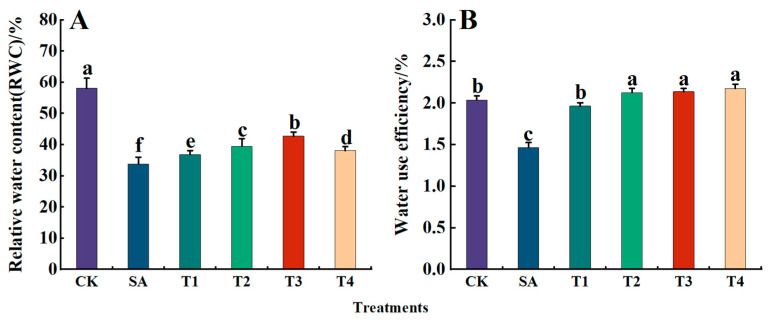
Effects of ZnO NPs on leaf relative water content and water use efficiency of M9-T337 seedlings under saline alkali stress. CK: normal treatment (control); SA: 100 mmol/L NaCl + NaHCO_3_ saline alkali stress; T1: saline alkali stress +50 mg/L ZnO NPs; T2: saline alkali stress +100 mg/L ZnO NPs; T3: saline alkali stress +150 mg/L ZnO NPs; T4: saline alkali stress +200 mg/L ZnO NPs. (**A**) RWC; (**B**) water use efficiency. Vertical bars represent the standard errors of the means of three replicates. Data show the mean ± SE (n = 3). Different lowercase letters indicate significant differences between treatments with *p* < 0.05.

**Figure 9 plants-14-00233-f009:**
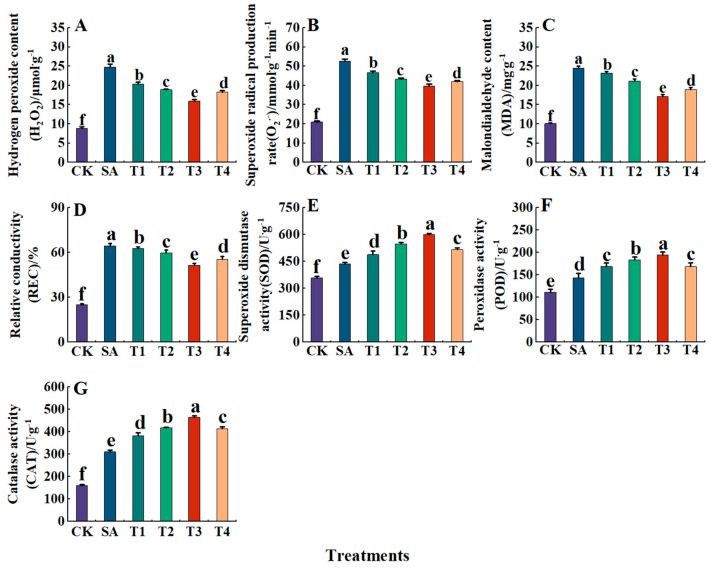
Effect of ZnO NPs on the degree of membrane lipid oxidation in M9-T337 seedlings under saline stress. CK: normal treatment (control); SA: 100 mmol/L NaCl + NaHCO_3_ saline alkali stress; T1: saline alkali stress +50 mg/L ZnO NPs; T2: saline alkali stress +100 mg/L ZnO NPs; T3: saline alkali stress +150 mg/L ZnO NPs; T4: saline alkali stress +200 mg/L ZnO NPs. (**A**) H_2_O_2_; (**B**) O_2_^•−^; (**C**) MDA; (**D**) REC; (**E**) SOD; (**F**) POD; (**G**) CAT. Vertical bars represent the standard errors of the means of three replicates. Data show the mean ± SE (n = 3). Different lowercase letters indicate significant differences between treatments with *p* < 0.05.

**Figure 10 plants-14-00233-f010:**
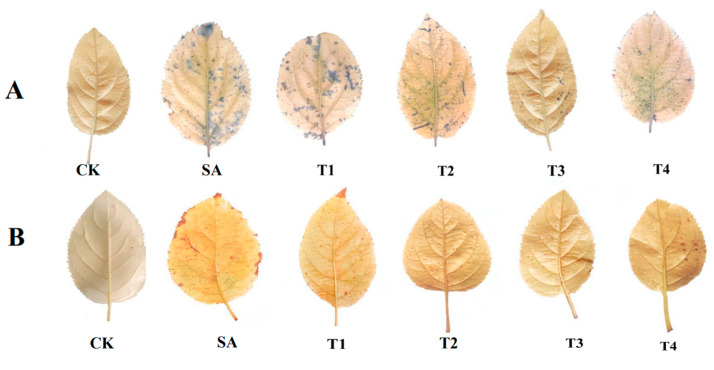
Effect of ZnO NPs on NBT and DAB staining of M9-T337 seedling leaves under salinity stress. CK: normal treatment (control); SA: 100 mmol/L NaCl + NaHCO_3_ saline alkali stress; T1: saline alkali stress +50 mg/L ZnO NPs; T2: saline alkali stress +100 mg/L ZnO NPs; T3: saline alkali stress +150 mg/L ZnO NPs; T4: saline alkali stress +200 mg/L ZnO NPs. (**A**) NBT staining; (**B**) DBA staining.

**Figure 11 plants-14-00233-f011:**
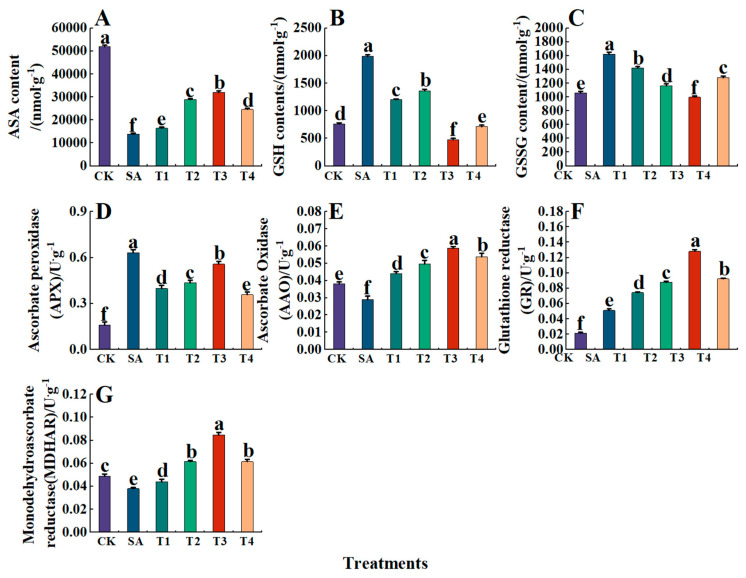
Effects of ZnO NPs on the activities of glutathione-cycling-related substances and enzymes in leaves of M9-T337 seedlings under saline alkali stress. CK: normal treatment (control); SA: 100 mmol/L NaCl + NaHCO_3_ saline alkali stress; T1: saline alkali stress +50 mg/L ZnO NPs; T2: saline alkali stress +100 mg/L ZnO NPs; T3: saline alkali stress +150 mg/L ZnO NPs; T4: saline alkali stress +200 mg/L ZnO NPs. (**A**) ASA; (**B**) GSH; (**C**) GSSG; (**D**) APX; (**E**) AAO; (**F**) GR; (**G**) MDHAR. Vertical bars represent the standard errors of the means three replicates. Data show the mean ± SE (n = 3). Different lowercase letters indicate significant differences between treatments with *p* < 0.05.

**Figure 12 plants-14-00233-f012:**
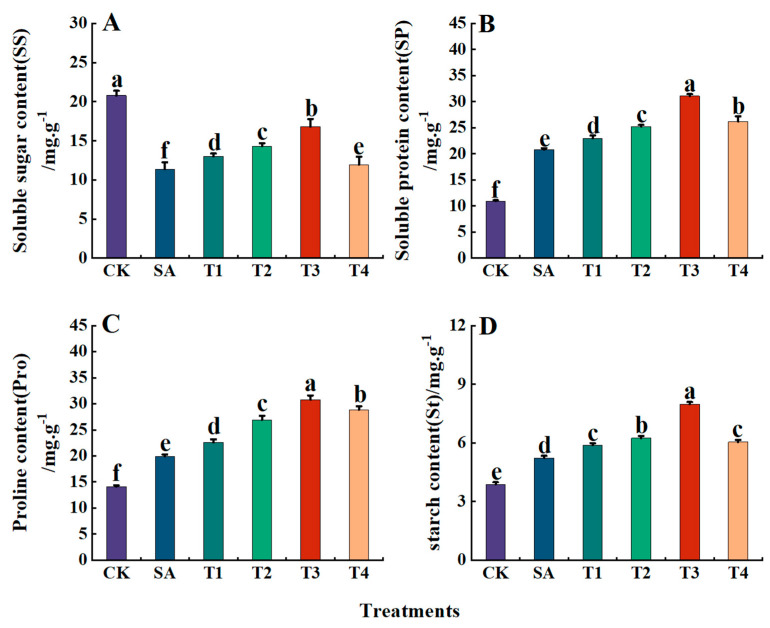
Determination of osmoregulatory substances in leaves of M9-T337 seedlings under saline alkali stress by ZnO NPs. CK: normal treatment (control); SA: 100 mmol/L NaCl + NaHCO_3_ saline alkali stress; T1: saline alkali stress +50 mg/L ZnO NPs; T2: saline alkali stress +100 mg/L ZnO NPs; T3: saline alkali stress +150 mg/L ZnO NPs; T4: saline alkali stress +200 mg/L ZnO NPs. (**A**) SS content; (**B**) SP content; (**C**) Pro content; (**D**) St content. Vertical bars represent the standard errors of the means of three replicates. Data show the mean ± SE (n = 3). Different lowercase letters indicate significant differences between treatments with *p* < 0.05.

**Figure 13 plants-14-00233-f013:**
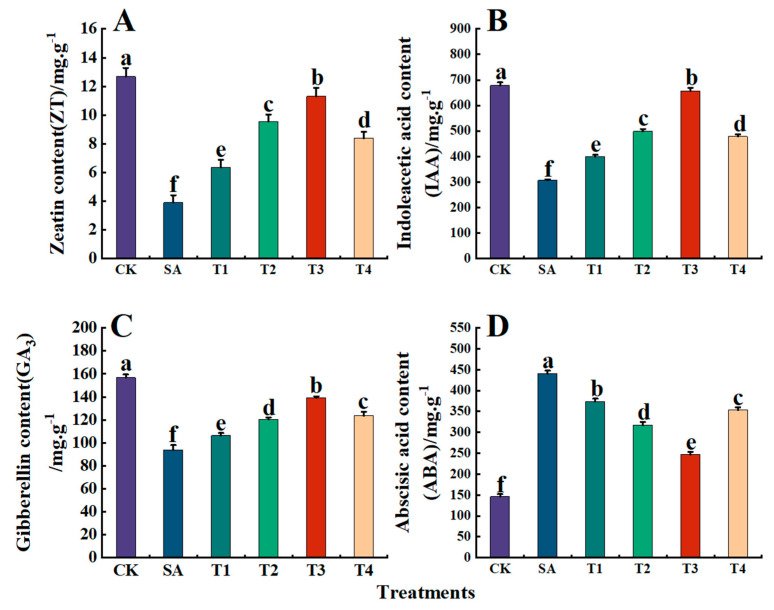
Effects of ZnO NPs on the contents of endogenous hormones in leaves of M9-T337 seedlings under saline alkali stress. CK: normal treatment (control); SA: 100 mmol/L NaCl + NaHCO_3_ saline alkali stress; T1: saline alkali stress +50 mg/L ZnO NPs; T2: saline alkali stress +100 mg/L ZnO NPs; T3: saline alkali stress +150 mg/L ZnO NPs; T4: saline alkali stress +200 mg/L ZnO NPs. (**A**) ZT content; (**B**) IAA content; (**C**) GA_3_ content; (**D**) ABA content. Vertical bars represent the standard errors of the means of three replicates. Data show the mean ± SE (n = 3). Different lowercase letters indicate significant differences between treatments with *p* < 0.05.

**Figure 14 plants-14-00233-f014:**
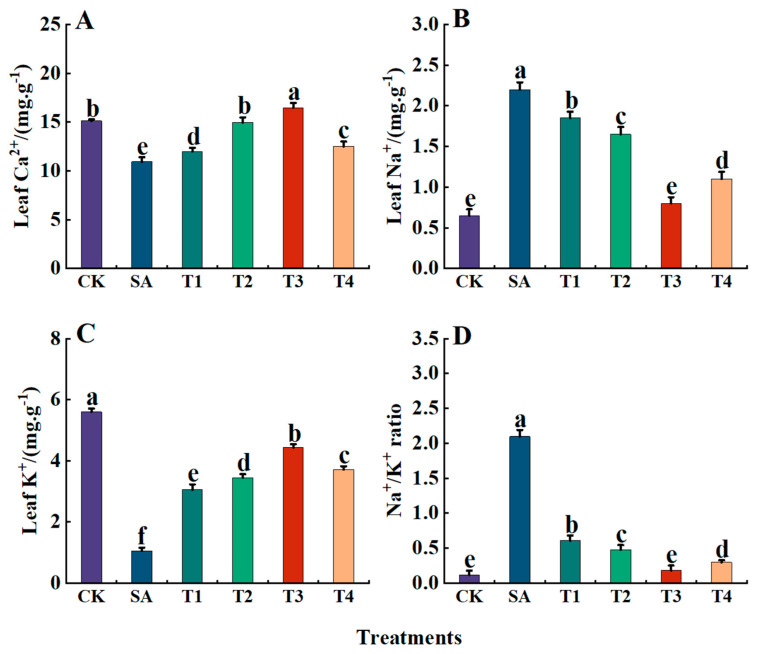
Effects of ZnO NPs on Ca^2+^, Na^+^ and K^+^ion contents and Na^+^/K^+^ in leaves of M9-T337 seedlings under saline alkali stress. CK: normal treatment (control); SA: 100 mmol/L NaCl + NaHCO_3_ saline alkali stress; T1: saline alkali stress +50 mg/L ZnO NPs; T2: saline alkali stress +100 mg/L ZnO NPs; T3: saline alkali stress +150 mg/L ZnO NPs; T4: saline alkali stress +200 mg/L ZnO NPs. (**A**) Ca^2+^; (**B**) Na^+^; (**C**) K^+^; (**D**) Na^+^/K^+^. Vertical bars represent the standard errors of the means of three replicates. Data show the mean ± SE (n = 3). Different lowercase letters indicate significant differences between treatments with *p* < 0.05.

**Figure 15 plants-14-00233-f015:**
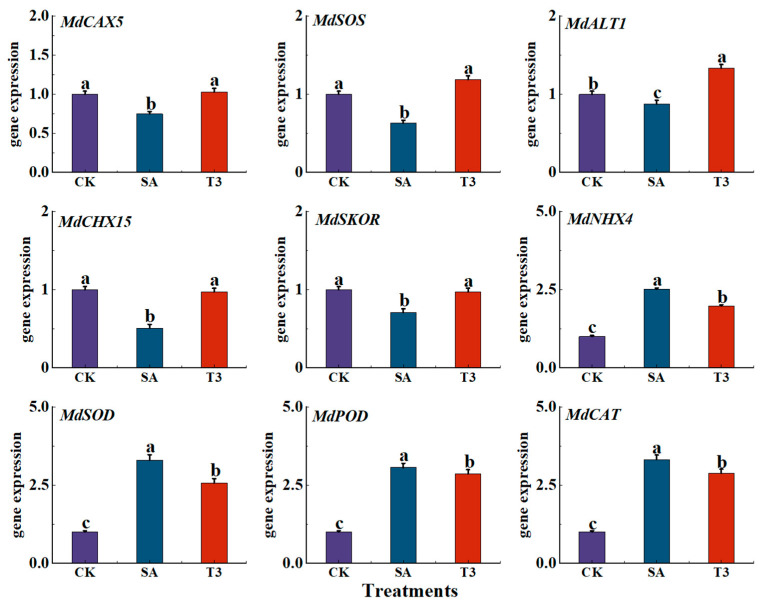
Effect of ZnO NPs on saline alkali (NaCl + NaHCO_3_) response gene expression in M9-T337 seedlings. CK: normal treatment (control); SA: 100 mmol/L NaCl + NaHCO_3_ saline alkali stress; T3: saline alkali stress +150 mg/L ZnO NPs. Vertical bars represent the standard errors of the means of three replicates. Data show the mean ± SE (n = 3). Different lowercase letters indicate significant differences between treatments with *p* < 0.05.

**Figure 16 plants-14-00233-f016:**
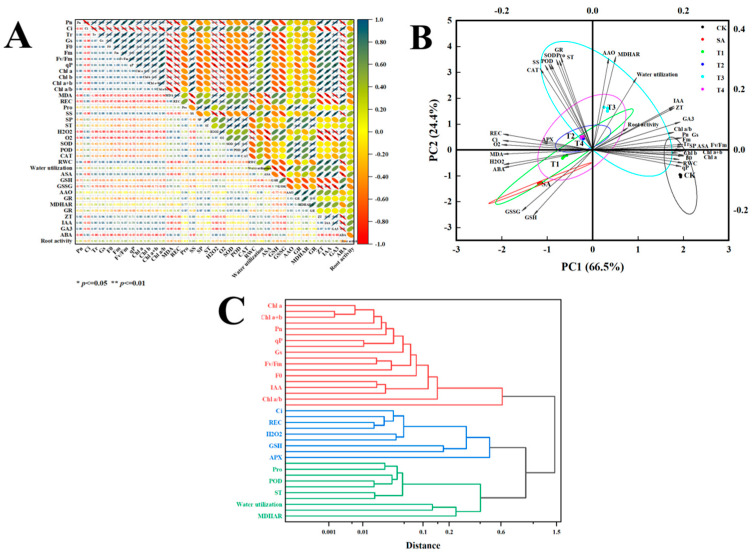
Correlation analysis, principal component analysis and cluster analysis of the indices of M9-T337 seedlings under different treatments, different symbols represent significant differences between treatments (* *p* < 0.05, ** *p* < 0.01). (**A**) Correlation analysis chart, (**B**) principal component analysis chart, and (**C**) cluster analysis chart.

**Table 1 plants-14-00233-t001:** Information of each gene and primer sequence used in qRT-PCR.

Gene Name	Full Gene Name	Gene Number (ID)	Forward Primer (5′-3′)	Reverse Primer (5′-3′)	Reference
*MdSKOR1*	Outwardly rectified K^+^ channel	LOC103452876	CATCCTGACAACTGGTGGTATCG	AAGTACCTCAGAGCAATCCGTTT	[48]
*MdNHX4*	Na^+^/H^+^ reverse transporter protein	LOC109230819	ACGAAACTCCTTTACTATACAGCCT	TGATACCACAGATAAGTGAGCATAG	[49]
*MdALT1*	Alanine transaminase	LOC103449816	GTTGTTTTTTTGCTCTGACGACT	GTTGAACCACAAAACCCTGCT	[48]
*MdSOS1*	Plasma membrane Na^+^/H^+^ reverse transporter protein	LOC103435911	TACATCATTTCTGGTATATCTTGTG	CAAGATGAAAATTAAGGTATTAGCA	[50]
*MdCHX15*	Cation/H(^+^) antiporter	LOC109229266	CCTCTTGGTACAGCATTGATAAAAA	GTTTGAACTTAATTTTGCAGCACA	[51]
*MdSOD*	Superoxide dismutase	LOC103453882	GTTCGATCCATCAAACGCCG	ATGAAGTCCAGGCTTGAGGC	[52]
*MdCAT*	Catalase	LOC103412104	CAACAACTTCCCGGTCTTC	GTAGGGCTTTCCGGATTTG	[53]
*MdPOD*	Peroxidase	LOC103452368	TGAAGAAACTGCTGGGGGTC	AACGTCCAAATTCGCATTGAT	[54]
*MdCAX5*	Ca^2+^/H^+^ reverse transporter protein	LOC109229266	AAAGTGTGAAGAAAGAGAAAATGC	AGTTCATTTGCTTGACATTCTTT	[55]

**Table 2 plants-14-00233-t002:** Effect of zinc oxide nanoparticles on total root length, root surface area, average root diameter, total root volume, root tip number and root vigor of M9-T337 seedlings under saline alkali stress.

Treatment	Total Root Length (mm)	Total Root Surface Area (cm^2^)	Average Root Diameter (mm)	Total Root Volume (mm^3^)	Root Tips Number	Root Vigor
CK	6637.17 ± 13.43 ^b^	7555.16 ± 13.30 ^b^	0.48 ± 0.05 ^cd^	2143.54 ± 17.78 ^c^	2571 ± 89.06 ^b^	62.88 ± 2.78 ^g^
SA	3758.49 ± 21.40 ^d^	4831.41 ± 21.67 ^d^	0.40 ± 0.02 ^d^	1173.40 ± 7.42 ^d^	2048 ± 19.39 ^c^	61.36 ± 3.01 ^d^
T1	5111.83 ± 12.04 ^c^	7265.36 ± 19.09 ^b^	0.53 ± 0.08 ^c^	2614.40 ± 19.79 ^b^	2168 ± 37.59 ^bc^	57.10 ± 1.88 ^f^
T2	6059.79 ± 13.12 ^b^	7688.33 ± 27.31 ^b^	0.75 ± 0.03 ^b^	1775.97 ± 16.31 ^c^	2781 ± 24.87 ^b^	55.14 ± 2.81 ^c^
T3	7415.43 ± 20.86 ^a^	9318.78 ± 15.05 ^a^	0.65 ± 0.03 ^a^	3241.24 ± 3.51 ^a^	4451 ± 21.54 ^a^	67.67 ± 3.72 ^e^
T4	5767.26 ± 16.59 ^b^	6830.22 ± 18.45 ^c^	0.55 ± 0.02 ^c^	1356.43 ± 24.74 ^d^	4284 ± 26.31 ^a^	58.67 ± 2.91 ^g^

Note: CK: normal treatment (control); SA: 100 mmol/L NaCl + NaHCO_3_ saline alkali stress; T1: saline alkali stress +50 mg/L ZnO NPs; T2: saline alkali stress +100 mg/L ZnO NPs; T3: saline alkali stress +150 mg/L ZnO NPs; T4: saline alkali stress +200 mg/L ZnO NPs. Error bars represent the standard deviation (±s.d.) for three replicates. Different lowercase letters in the same column indicate statistically significant differences (*p* < 0.05).

**Table 3 plants-14-00233-t003:** Effect of ZnO NPs on aboveground and root dry (fresh) weight of M9-T337 leaf under saline stress.

Treatment	Fresh Weight Biomass (g·Plant^−1^ FW)			Dry Weight Biomass (g·Plant^−1^ DW)		
	Plant	Shoot	Root	Plant	Shoot	Root
CK	9.41 ± 0.23 ^g^	5.06 ± 0.22 ^g^	4.35 ± 0.35 ^g^	2.94 ± 0.24 ^c^	1.91 ± 0.25 ^g^	1.04 ± 0.02 ^f^
SA	5.44 ± 0.31 ^e^	3.01 ± 0.27 ^e^	2.43 ± 0.29 ^e^	1.96 ± 0.09 ^g^	1.23 ± 0.24 ^d^	0.73 ± 0.04 ^e^
T1	6.86 ± 0.21 ^f^	3.65 ± 0.26 ^c^	3.21 ± 0.22 ^f^	2.21 ± 0.18 ^f^	1.41 ± 0.18 ^c^	0.79 ± 0.06 ^c^
T2	8.95 ± 0.26 ^a^	5.01 ± 0.31 ^b^	4.24 ± 0.23 ^e^	3.41 ± 0.21 ^c^	2.57 ± 0.17 ^b^	1.07 ± 0.04 ^d^
T3	11.85 ± 0.29 ^d^	4.78 ± 0.23 ^e^	7.07 ± 0.14 ^d^	3.88 ± 0.17 ^d^	2.07 ± 0.24 ^f^	1.81 ± 0.01 ^f^
T4	8.55 ± 0.33 ^c^	4.71 ± 0.42 ^d^	3.55 ± 0.26 ^c^	2.97 ± 0.15 ^e^	1.91 ± 0.27 ^c^	0.85 ± 0.03 ^b^

Note: CK: normal treatment (control); SA: 100 mmol/L NaCl + NaHCO_3_ saline alkali stress; T1: saline alkali stress +50 mg/L ZnO NPs; T2: saline alkali stress +100 mg/L ZnO NPs; T3: saline alkali stress +150 mg/L ZnO NPs; T4: saline alkali stress +200 mg/L ZnO NPs. Error bars represent the standard deviation (±s.d.) for three replicates. Different lowercase letters in the same column indicate statistically significant differences (*p* < 0.05).

**Table 4 plants-14-00233-t004:** Effects of ZnO NPs on stomatal parameters of leaves of M9-T337 seedlings under saline alkali stress.

Treatments	Lp (μm)	Wp (μm)	SD (mm^2^)	SS (μm^2^)	Sa (μm^2^)	Gsmax (mol m^−2^ s^−2^)
CK	15.13 ± 0.35 ^a^	2.35 ± 0.45 ^a^	220.33 ± 2.52 ^b^	26.16 ± 6.53 ^a^	294.94 ± 10.56 ^a^	0.63 ± 0.06 ^b^
SA	9.33 ± 0.91 ^b^	0.63 ± 0.077 ^f^	123.67 ± 9.71 ^e^	5.78 ± 0.70 ^c^	263.09 ± 28.89 ^a^	0.43 ± 0.04 ^b^
T1	10.73 ± 0.66 ^c^	0.89 ± 0.099 ^df^	173.67 ± 9.50 ^d^	7.85 ± 0.79 ^c^	277.24 ± 73.74 ^a^	0.62 ± 0.04 ^b^
T2	11.77 ± 0.45 ^b^	1.14 ± 0.11 ^cd^	205.00 ± 9.64 ^c^	12.19 ± 0.89 ^bc^	354.51 ± 68.87 ^a^	0.61 ± 0.11 ^b^
T3	14.57 ± 0.59 ^a^	1.82 ± 0.17 ^b^	266.00 ± 9.00 ^a^	16.51 ± 4.51 ^b^	266.38 ± 41.57 ^a^	0.96 ± 0.22 ^a^
T4	12.5 ± 0.19 ^b^	1.48 ± 0.07 ^bc^	223.33 ± 8.02 ^b^	11.72 ± 1.55 ^bc^	265.38 ± 48.76 ^a^	0.90 ± 0.16 ^a^

Note: CK: normal treatment (control); SA: 100 mmol/L NaCl + NaHCO_3_ saline alkali stress; T1: saline alkali stress +50 mg/L ZnO NPs; T2: saline alkali stress +100 mg/L ZnO NPs; T3: saline alkali stress +150 mg/L ZnO NPs; T4: saline alkali stress +200 mg/L ZnO NPs. Error bars represent the standard deviation (±s.d.) for three replicates. Different lowercase letters in the same column indicate statistically significant differences (*p* < 0.05).

## Data Availability

Data will be made available on request. The data are not publicly available due to lab’s policies and confidentiality agreements.

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
