# Peer review of "Nano-Zinc Oxide Can Enhance the Tolerance of Apple Rootstock M9-T337 Seedlings to Saline Alkali Stress by Initiating a Variety of Physiological and Biochemical Pathways"

_plants, 2025, doi:10.3390/plants14020233_

Round 1

Reviewer 1 Report

Comments and Suggestions for Authors

Soil salinization severely restricts the growth and development of crops globally and is becoming a severe problem for agriculture all over the world. The current research targeted an important subject, namely the tolerance and adaptability of apple rootstocks to saline conditions. The research is well designed, but is not clearly described: e.g. frequency of irrigation/treatments applied to the plants? T1, T2, T3 and T4 have no NaCl added? If it is so, how dis you test the role/efficiency of ZnO NPs? Several plant names are incorrectly correlated to their latin names through the text early in the introduction part. The main aim of the research is clearly stated, however the justification on treatment selection and the presentation of the results are a little bit shuffling. There are important data which loose its significance through the incomplete presentation. The graphs should be magnified for a better understanding. The discussion part should focus more on the results obtained and their importance for future studies and practical applications. Further suggestions could be found in the attachment file.

Author Response

Dear Editors and Reviewers,

Thank you for your kind work and for the reviewers’ comments concerning our manuscript entitled “Nano-zinc oxide can enhance the tolerance of apple rootstock M9-T337 seedlings to saline-alkali stress by initiating a variety of physiological and biochemical pathways”. Those comments are all valuable and very helpful for revising and improving our paper, as well as the important guiding significance to our researches. We have studied comments carefully and have made correction. Revised portion are marked in red in the paper. The main corrections in the paper and the responses to reviewers’comments are asas follows:

Comments to the Author

Soil salinization severely restricts the growth and development of crops globally and is becoming a severe problem for agriculture all over the world. The current research targeted an important subject, namely the tolerance and adaptability of apple rootstocks to saline conditions. The research is well designed, but is not clearly described:

  1. frequency of irrigation/treatments applied to the plants?

Response: Thank you very much for reviewers’ valuable advice, First, different concentrations of ZnO NPs were pretreated, except for CK and SA, T1-T4 treatments were equally poured with 200 mL of ZnO NPs suspension with different concentrations, once every three days for five times. After the last ZnO NPs treatment for seven days, the saline-alkali stress treatment was carried out, except CK, 200 ml saline-alkali solution (100 mmol/L NaCl + NaHCO3) was poured into SA - T4 treatments every seven days for three times. I have added accordingly in the original text, as detailed in manuscript line 172-177.

  1. T1, T2, T3 and T4 have no NaCl added? If it is so, how dis you test the role/efficiency of ZnO NPs?

Response: Thank you very much for reviewers’ valuable advice,the T1-T4 treatments were equal watering of 200 mL of equal amounts of suspensions of ZnO NPs of different concentrations on the basis of saline and alkaline stresses, which I have added accordingly in the original text, as detailed in manuscript line 172-177.

  1. Line 25, “soluble protein (SS)”shouldn't be SP?

Response: Thank you very much for reviewers’ valuable advice, “SS” have be changed to “SP”. I have added accordingly in the original text, as detailed in manuscript line 25.

  1. Line 25, “soluble sugar (SP)”shouldn't be SS?

Response: Thank you very much for reviewers’ valuable advice, “SP” have be changed to “SS”. I have added accordingly in the original text, as detailed in manuscript line 25.

  1. Line 89-91,“Studies have found that nanoparticles (NPs) can act on plants through leaf spraying and root application of plants, Which are eventually distributed in various organs of plants through absorption and transport of plants.” This sentence should be revised to avoid any confusion.

Response: Thank you very much for reviewers’ valuable advice, this sentence has been rewritten: Studies have found that leaf spraying or root application of nanoparticles(NPs) can be absorbed and utilized by the plant, and ultimately distributed in all organs of the plant. I have added accordingly in the original text, as detailed in manuscript line 89-91.

  1. eggplants are called Solanum melongena, while Phaseolus vulgaris are beans. Please check the validity of information according to the citation.

Response: Thank you very much for reviewers’ valuable advice, I have changed “Phaseolus vulgaris” to “Solanum melongena”. I have added accordingly in the original text, as detailed in manuscript line 112.

  1. “Lycium barbarum” should be italic.

Response: Thank you very much for reviewers’ valuable advice, I have italicized “Lycium barbarum”. I have added accordingly in the original text, as detailed in manuscript line 114.

  1. cabbage is Brassica oleracea while Brassica napus is rapeseed! Check the information according to the citation!

Response: Thank you very much for reviewers’ valuable advice, I have changed “Brassica napus” to “Brassica oleracea”. I have added accordingly in the original text, as detailed in manuscript line 110.

  1. “sorghum(Sorghum bicolor)” space should be added between

Response: Thank you very much for reviewers’ valuable advice, I have added spaces. I have added accordingly in the original text, as detailed in manuscript line 115.

  1. Line 116, 122, 125, 129, 195, 197, 211, 215, 226, 636, and 651 check the journal's instructions regarding citation methods.

Response: Thank you very much for reviewers’ valuable advice, I have unified the manuscripts. I have changed the Türkoğlu A et al., Lalarukh et al., Azmat et al., Dang K et al., Arnon et al., Liu et al.,(2022), Lin et al.,(2022), Yuan et al(2013)., Zheng et al.,(2018), Kasivelu et al., Ayari et al., to Türkoğlu et al, Lalarukh et al, Azmat et al, Dang et al, Liu et al, Lin et al, Yuan et al, Zheng et al, Kasivelu et al, Ayari et al. I have added accordingly in the original text, as detailed in manuscript line 116, 122, 125, 129, 195, 197, 211, 215, 226, 636, and 651.

  1. should be Hoagland not Hoangland!

Response: Thank you very much for reviewers’ valuable advice, I have changed Hoangland to Hoagland. I have added accordingly in the original text, as detailed in manuscript line171.

  1. “After ZnO NPs treatment, The above index values have increased, and the effect of T3 treatment was the most obvious.”correct typing mistakes.

Response: Thank you very much for reviewers’ valuable advice, the sentence has been reorganised in language such as: After treatment with ZnO NPs, the values of all the above indexes were significantly increased, with the highest values being reached under the T3 treatment. I have added accordingly in the original text, as detailed in manuscript line 286-288.

  1. “The results of this experiment showed that the accumulation of biomass, Chl a, Chl b, Chl a/b and Chl a+b in the leaves of ZnO NPs treated plants were significantly higher than those treated with saline-alkali stress alone, which was consistent with the study of Raliya et al. in maize.” no plants can be treated woth stress, eventually subjected to saline-alkali stress.

Response: Thank you very much for reviewers’ valuable advice, this sentence has been rewritten: The results of this experiment showed that the accumulation of biomass, Chl a, Chl b, Chl a/b and Chl a+b were significantly higher in plants treated with ZnO NPs under saline-alkali stress than in plants treated with saline-alkali stress alone. I have added accordingly in the original text, as detailed in manuscript line 630-632.

Reviewer 2 Report

Comments and Suggestions for Authors

Dear Authors,

The submitted manuscript (plants-3391423) is an instant molecular-level study based on the investigation of candidate genes regulating and enhancing the tolerance of apple rootstock M9-T337 seedlings to saline-alkali stress by initiating a variety of physiological and biochemical pathways. The authors performed relatively experimental analysis but some revisions for further refinement.

* The English could be extensively improved to more clearly express the research.

* The introduction should be written in proper scientific style by adding a relevant literature review on molecular and genetic regulation of tolerant plants grown under saline-alkali stress effects in apples and other crop plants, respectively.

* Use the full name of abbreviations when writing for the first time; then you can use the abbreviations in the entire text.

* Paragraphs of written text are presented in irregular format. Please rearrange the paragraphs in proper format, and each paragraph should be separated and contain a maximum of 10-12 lines.

* Table 1. Please complete the table with other molecular information of genes, along with adding the column of referenced publication. Also, add the information in the text about how you identified those genes.

* Figure 1. Label the image (a) and graph (b) for more clarification. Please add the complete caption in the title.

* Table 3. Please add the complete caption in the title. Add the details of statistical letters, and statistical letters should be added in superscript format.

* Figure 4. Please label the images of CK, SA, T1, T2, T3, and T4, respectively.

* Table 4. The statistical letters should be added in superscript format.

* Figure 10. Please label and enlarge the photos of seedling leaves under salinity stress.

* Figures 5, 6, 7, 8, 9, 11, 12, 13, and 14. The font size should be enlarged enough for the reader’s availability.

* Figures 15 and 16 should be merged into the same figure.

* In discussion, there is a strong need to add the results of more comparative studies explaining the physiological and genetic regulation of the tolerance trait in apples, respectively.

* In discussion, please rearrange the paragraphs in proper format, and each paragraph should be separated and contain a maximum of 10-12 lines.

Comments on the Quality of English Language

The English could be extensively improved to more clearly express the research.

Author Response

Dear Editors and Reviewers,

Thank you for your kind work and for the reviewers’ comments concerning our manuscript entitled “Nano-zinc oxide can enhance the tolerance of apple rootstock M9-T337 seedlings to saline-alkali stress by initiating a variety of physiological and biochemical pathways”. Those comments are all valuable and very helpful for revising and improving our paper, as well as the important guiding significance to our researches. We have studied comments carefully and have made correction. Revised portion are marked in red in the paper. The main corrections in the paper and the responses to reviewers’comments are asas follows:

Comments to the Author

The submitted manuscript (plants-3391423) is an instant molecular-level study based on the investigation of candidate genes regulating and enhancing the tolerance of apple rootstock M9-T337 seedlings to saline-alkali stress by initiating a variety of physiological and biochemical pathways. The authors performed relatively experimental analysis but some revisions for further refinement.

  1. The English could be extensively improved to more clearly express the research.

Response: Thank you very much for reviewers’ valuable advice, I have corrected the misrepresentation of English in the manuscript. I have added accordingly in the original text, as detailed in manuscript line 89-91, line167-172, lin684-694, and line744-756.

  1. The introduction should be written in proper scientific style by adding a relevant literature review on molecular and genetic regulation of tolerant plants grown under saline-alkali stress effects in apples and other crop plants, respectively.

Response: Thank you very much for reviewers’ valuable advice, I have added in the introduction the study of molecular and genetic regulation of salt-tolerant plants of apples and other crops grown under the influence of saline stress. I have added accordingly in the original text, as detailed in manuscript line 56-84.

Such as: During plant growth and evolution, salt-tolerant plants are apparently able to adapt to environmental changes through a complex set of molecular and genetic regulatory mechanisms. Specifically, the ion regulatory pathway, plants rely primarily on ion transport proteins (Na⁺/H⁺ reverse transporter proteins and K⁺ channels) in the cell membrane to maintain the intracellular balance of Na⁺/K⁺ ratios, thereby prevent ion toxicity.4 In addition, salt-tolerant plants effectively scavenge excess reactive oxygen species (ROS) by increasing the activity of antioxidant enzymes (SOD, POD, CAT) and accumulating non-enzymatic antioxidants (APX, AAO, GR, and MDHAR), thereby protecting cells from oxidative damage. At the same time, plants increase cellular osmotic pressure and thus maintain the stability of cellular structure and water balance by accumulating osmoregulatory substances, such as Pro, SS and GB; phytohormones(including ABA, C2H4 and GA3) play a key role in regulating the plant's adaptation to saline and alkaline stresses, and they can enhance the plant's salinity tolerance by affecting the pathways of growth, stomatal opening and closing and the synthesis of osmoregulatory substances. Salt tolerance is the result of synergistic expression of multiple genes involving multiple signaling pathways, including the abscisic acid pathway, the SOS pathway, and the protein kinase pathway. Among them, in the SOS pathway, the SOS1, SOS2, and SOS3 genes act together to maintain intracellular ionic homeostasis by regulating Na⁺/K⁺ balance, thereby enhancing salt tolerance in plants. In addition, transcription factors such as the MYB family regulate the synthesis of antioxidants and the transport of ions by modulating the expression of downstream genes, further improving the salt tolerance of plants; in the abscisic acid pathway, ABA binds to the receptor and inhibits PP2C, activating the SnRK2 kinase, which in turn phosphorylates the transcription factors(such as bZIP), and regulates the expression of salt tolerance genes.8 Notably, recent studies have revealed that the receptor-like kinase FERONIA affects plant salt tolerance by regulating the metabolic flow of the photorespiratory pathway, providing a new perspective for understanding the mechanism of plant salt tolerance. These complex regulatory mechanisms interact with each other and together enable salt-tolerant plants to survive and grow under saline stress.

  1. Use the full name of abbreviations when writing for the first time; then you can use the abbreviations in the entire text.

Response: Thank you very much for reviewers’ valuable advice, I have rewritten the first indicator of the full text into a full name, and the indicators that appear later are rewritten as abbreviations. Such as: line222-223: Pro, SS, SP;line 226: IAA, ABA, GA3, and ZT; line 239: O2.-; line 242: H2O2; line 364-369 and line397-398: Chl a, Chl b, Chl a+b, and Chl a/b; line 405: Pn, Ci, Gs, Tr;line 412: Fm, F0, Fv/Fm, qP; line 417 and line 451: RWC; line 427-428: H2O2, O2.-, MDA, REC;line 460-461: H2O2, O2.-, MDA, REC, SOD, POD, CAT; line 519-520: SS, SP, St, Pro; line 540: ZT, IAA, GA3, ABA; line 561: Ca2+, Na+, K+, Na+/K+.

  1. Table 1. Please complete the table with other molecular information of genes, along with adding the column of referenced publication. Also, add the information in the text about how you identified those genes.

Response: Thank you very much for reviewers’ valuable advice, I have filled in Table 1 with additional molecular information for the genes and added a reference column. Information on how these genes were identified has also been added to the text. I have added accordingly in the original text, as detailed in manuscript line 249-261.

Such as: Total RNA was extracted using an RNA extraction kit from Bi-oTeke, Beijing, China, and reverse transcription was performed using TaKaRa's PrimeScriptTM RT kit and gDNA Eraser (Perfect Real Time). The primers were designed by Shanghai Sangong Bioengineering Co., Ltd. and the sequences were obtained from NCBI database. The real-time quantitative PCR was performed using apple cDNA as template and GAPDH as reference. The reaction system consisted of TB GreenTM Premix Ex Taq II, 10 μmol each of upstream and downstream primers, cDNA template and dd H2O in 10, 1, 1, 2 and 6 μL, respectively, and the reaction conditions included pre-denaturation at 95 ℃ for 3 min, followed by denaturation at 95 ℃ for 5 s, annealing at 56 ℃ for 30 s, and extension at 72 ℃ for 30 s. Forty cycles were repeated. Each sample was triple-treated to ensure the reliability of the results. The data obtained from quantification were analyzed by 2-∆∆CT method.

Table 1. Information of each gene and primer sequences used in qRT-PCR.

Gene name

Full gene name

gene number(ID)

Forward primer (5′-3′)

Reverse primer (5′-3′)

reference

MdSKOR1

Outwardly rectified K+ channel

LOC103452876

CATCCTGACAACTGGTGGTATCG

AAGTACCTCAGAGCAATCCGTTT

48

MdNHX4

Na+/H+ reverse transporter protein

LOC109230819

ACGAAACTCCTTTACTATACAGCCT

TGATACCACAGATAAGTGAGCATAG

49

MdALT1

alanine transaminase

LOC103449816

GTTGTTTTTTTGCTCTGACGACT

GTTGAACCACAAAACCCTGCT

48

MdSOS1

Plasma membrane Na+/H+ reverse transporter protein

LOC103435911

TACATCATTTCTGGTATATCTTGTG

CAAGATGAAAATTAAGGTATTAGCA

50

MdCHX15

cation/H(+) antiporter 

LOC109229266

CCTCTTGGTACAGCATTGATAAAAA

GTTTGAACTTAATTTTGCAGCACA

51

MdSOD

superoxide dismutase

LOC103453882

GTTCGATCCATCAAACGCCG

ATGAAGTCCAGGCTTGAGGC

52

MdCAT

catalase

LOC103412104

CAACAACTTCCCGGTCTTC

GTAGGGCTTTCCGGATTTG

53

MdPOD

peroxidase

LOC103452368

TGAAGAAACTGCTGGGGGTC

AACGTCCAAATTCGCATTGAT

54

MdCAX5

Ca2+/H+ reverse transporter protein

LOC109229266

AAAGTGTGAAGAAAGAGAAAATGC

AGTTCATTTGCTTGACATTCTTT

55

  1. Figure 1. Label the image (a) and graph (b) for more clarification. Please add the complete caption in the title.

Response: Thank you very much for reviewers’ valuable advice, I have labeled the picture (a) and the chart (b) and the title has been modified from Figure.1 Characterization of ZnO NPs to Figure.1 Transmission electron microscopy (TEM) imaging and TEM grain size measurements of ZnO NPs. I have added accordingly in the original text, as detailed in manuscript line 274-276.

  1. Table 3. Please add the complete caption in the title. Add the details of statistical letters, and statistical letters should be added in superscript format.

Response: Thank you very much for reviewers’ valuable advice, I have added the complete caption in the title, the details of statistical letters and added superscript formatting to statistical letters. I have added accordingly in the original text, as detailed in manuscript line 323-324 and line 327-329. Such as:

Table 3. Effect of ZnO NPs on aboveground and root dry (fresh) weight of M9-T337 leaf under saline stress.;

(2)

treatment

Fresh weight biomass(g.plant-1 FW)

Dry weight biomass(g.plant-1 DW)

Plant

Shoot

Root

Plant

Shoot

Root

CK

9.41±0.23g

5.06±0.22g

4.35±0.35g

2.94±0.24c

1.91±0.25g

1.04±0.02f

SA

5.44±0.31e

3.01±0.27e

2.43±0.29e

1.96±0.09g

1.23±0.24d

0.73±0.04e

T1

6.86±0.21f

3.65±0.26c

3.21±0.22f

2.21±0.18f

1.41±0.18c

0.79±0.06c

T2

8.95±0.26a

5.01±0.31b

4.24±0.23e

3.41±0.21c

2.57±0.17b

1.07±0.04d

T3

11.85±0.29d

4.78±0.23e

7.07±0.14d

3.88±0.17d

2.07±0.24f

1.81±0.01f

T4

8.55±0.33c

4.71±0.42d

3.55±0.26c

2.97±0.15e

1.91±0.27c

0.85±0.03b

Error bars represent the standard deviation (±s.d.) for three replicates. Different lowercase letters in the same column indicate statistically significant differences (P<0.05).

  1. Figure 4. Please label the images of CK, SA, T1, T2, T3, and T4, respectively.

Response: Thank you very much for reviewers’ valuable advice, I have relabeled the images of CK, SA, T1, T2, T3 and T4. I have added accordingly in the original text, as detailed in manuscript line 351.

  1. Table 4. The statistical letters should be added in superscript format.

Response: Thank you very much for reviewers’ valuable advice, I have added superscript formatting to the stat letters. I have added accordingly in the original text, as detailed in manuscript line 358-362.

Such as:

Treatments

Lp(μm)

Wp(μm)

SD(mm2)

SS(μm2)

Sa(μm2)

Gsmax(mol m-2 s-2)

CK

15.13±0.35a

2.35±0.45a

220.33±2.52b

26.16±6.53a

294.94±10.56a

0.63±0.06b

SA

9.33±0.91b

0.63±0.077f

123.67±9.71e

5.78±0.70c

263.09±28.89a

0.43±0.04b

T1

10.73±0.66c

0.89±0.099df

173.67±9.50d

7.85±0.79c

277.24±73.74a

0.62±0.04b

T2

11.77±0.45b

1.14±0.11cd

205.00±9.64c

12.19±0.89bc

354.51±68.87a

0.61±0.11b

T3

14.57±0.59a

1.82±0.17b

266.00±9.00a

16.51±4.51b

266.38±41.57a

0.96±0.22a

T4

12.5±0.19b

1.48±0.07bc

223.33±8.02b

11.72±1.55bc

265.38±48.76a

0.90±0.16a

  1. Figure 10. Please label and enlarge the photos of seedling leaves under salinity stress.

Response: Thank you very much for reviewers’ valuable advice, I have labeled and enlarged photos of seedling leaves under salinity stress. I have added accordingly in the original text, as detailed in manuscript line 464.

  1. Figures 5, 6, 7, 8, 9, 11, 12, 13, and 14. The font size should be enlarged enough for the reader’s availability.

Response: Thank you very much for reviewers’ valuable advice, I have enlarged the font in images 5, 6, 7, 8, 9, 11, 12, 13, and 14. I have added accordingly in the original text, as detailed in manuscript line 394, 402, 408, 446, 456, 493, 515, 535 and 556.

  1. Figures 15 and 16 should be merged into the same figure.

Response: Thank you very much for reviewers’ valuable advice, I have merged figures 15 and 16 into a single figure. I have added accordingly in the original text, as detailed in manuscript line 614.

  1. In discussion, there is a strong need to add the results of more comparative studies explaining the physiological and genetic regulation of the tolerance trait in apples, respectively.

Response: Thank you very much for reviewers’ valuable advice, I have added more comparative findings in the discussion section and added a description of the physiological and genetic regulation of apple tolerance. I have added accordingly in the original text, as detailed in manuscript line 645-651, line 679-683, and line 723-727.

Such as: (1)Dang et al showed that under stress conditions, photosynthesis is limited in two major ways: stomatal limitation, which is due to a decrease in stomatal conductance, resulting in an insufficient concentration of intercellular CO2 to meet the demands of photosynthesis; and non-stomatal limitation, which is mainly due to a decrease in chloroplast activity as well as a decrease in the activity of ribulose-1,5-bisphosphate carboxylase (Rubisco) and reduced regeneration of ribulose-1,5-bisphosphate (RuBP); (2)Meanwhile, Chen et al found that ZnO NPs could alleviate the damage caused by salt stress on the PS II photosystem of tea tree by balancing the electron transfer from the donor side, acceptor side, and reaction centers of PS II photosystem in their study;73 Liu et al also demonstrated in spinach that the application of ZnO NPs significantly up-regulated the expression of light-harvesting complex II (LHCB)-related genes;(3)Meanwhile Prakash et al in rice showed that the rise in ASA, GSH content and related enzyme activities could be attributed to ZnONPs facilitating the conversion of AsA to DHA and the equilibrium exchange between GSH and GSSG, which maintains the redox homeostasis of the leaves and enhances their salinity tolerance.

  1. In discussion, please rearrange the paragraphs in proper format, and each paragraph should be separated and contain a maximum of 10-12 lines.

Response: Thank you very much for reviewers’ valuable advice, I have rearranged the paragraphs and incorporated the suggestions from two other reviewers to make a comprehensive revision to the discussion. Please refer to lines 621-771 of the manuscript for details.

Reviewer 3 Report

Comments and Suggestions for Authors

The manuscript is devoted to the urgent problem of increasing apple tree resistance to salt stress using zinc nanoparticles. The article is well written. The experimental material is sufficient. Modern research methods have been used. Illustrative material is clear, easy to understand. Tables are readable. The authors' conclusions are logical. The list of sources is sufficient. The manuscript makes a good impression. 

Some remarks to the authors.

1. It is necessary to expand the list of key words by listing the studied parameters.

2. In the Introduction, it is better to add information on the mechanisms of salt tolerance in plants.

3. In the Introduction, it would be good to emphasize more clearly the novelty of this particular study, as well as to explain why these particular physiological parameters were chosen for the study.

Author Response

Dear Editors and Reviewers,

Thank you for your kind work and for the reviewers’ comments concerning our manuscript entitled “Nano-zinc oxide can enhance the tolerance of apple rootstock M9-T337 seedlings to saline-alkali stress by initiating a variety of physiological and biochemical pathways”. Those comments are all valuable and very helpful for revising and improving our paper, as well as the important guiding significance to our researches. We have studied comments carefully and have made correction. Revised portion are marked in red in the paper. The main corrections in the paper and the responses to reviewers’comments are asas follows:

Comments to the Author

The manuscript is devoted to the urgent problem of increasing apple tree resistance to salt stress using zinc nanoparticles. The article is well written. The experimental material is sufficient. Modern research methods have been used. Illustrative material is clear, easy to understand. Tables are readable. The authors' conclusions are logical. The list of sources is sufficient. The manuscript makes a good impression. Some remarks to the authors.

  1. It is necessary to expand the list of key words by listing the studied parameters.

Response: Thank you very much for reviewers’ valuable advice, I have listed the parameters of the research to expand the list of keywords, I have added accordingly in the original text, as detailed in manuscript line 39.

Such as: Osmoregulation; ROS removal; Hormones.

  1. In the Introduction, it is better to add information on the mechanisms of salt tolerance in plants.

Response: Thank you very much for reviewers’ valuable advice, I have already added the mechanisms of salt tolerance in plants in the introduction. I have added accordingly in the original text, as detailed in manuscript line 56-84.

Such as: During plant growth and evolution, salt-tolerant plants are apparently able to adapt to environmental changes through a complex set of molecular and genetic regulatory mechanisms. Specifically, the ion regulatory pathway, plants rely primarily on ion transport proteins (Na⁺/H⁺ reverse transporter proteins and K⁺ channels) in the cell membrane to maintain the intracellular balance of Na⁺/K⁺ ratios, thereby prevent ion toxicity. In addition, salt-tolerant plants effectively scavenge excess reactive oxygen species (ROS) by increasing the activity of antioxidant enzymes (SOD, POD, CAT) and accumulating non-enzymatic antioxidants (APX, AAO, GR, and MDHAR), thereby protecting cells from oxidative damage. At the same time, plants increase cellular osmotic pressure and thus maintain the stability of cellular structure and water balance by accumulating osmoregulatory substances, such as Pro, SS and GB; phytohormones(including ABA, C2H4 and GA3) play a key role in regulating the plant's adaptation to saline and alkaline stresses, and they can enhance the plant's salinity tolerance by affecting the pathways of growth, stomatal opening and closing and the synthesis of osmoregulatory substances. Salt tolerance is the result of synergistic expression of multiple genes involving multiple signaling pathways, including the abscisic acid pathway, the SOS pathway, and the protein kinase pathway. Among them, in the SOS pathway, the SOS1, SOS2, and SOS3 genes act together to maintain intracellular ionic homeostasis by regulating Na⁺/K⁺ balance, thereby enhancing salt tolerance in plants. In addition, transcription factors such as the MYB family regulate the synthesis of antioxidants and the transport of ions by modulating the expression of downstream genes, further improving the salt tolerance of plants; in the abscisic acid pathway, ABA binds to the receptor and inhibits PP2C, activating the SnRK2 kinase, which in turn phosphorylates the transcription factors(such as bZIP), and regulates the expression of salt tolerance genes. Notably, recent studies have revealed that the receptor-like kinase FERONIA affects plant salt tolerance by regulating the metabolic flow of the photorespiratory pathway, providing a new perspective for understanding the mechanism of plant salt tolerance. These complex regulatory mechanisms interact with each other and together enable salt-tolerant plants to survive and grow under saline stress.

  1. In the Introduction, it would be good to emphasize more clearly the novelty of this particular study, as well as to explain why these particular physiological parameters were chosen for the study.

Response: Thank you very much for reviewers’ valuable advice, I have explained why the specific physiological parameters were chosen in the Introduction section(line56-69) and emphasized more clearly the novelty of this study(line 98-103).

Such as: (1) During plant growth and evolution, salt-tolerant plants are apparently able to adapt to environmental changes through a complex set of molecular and genetic regulatory mechanisms. Specifically, the ion regulatory pathway, plants rely primarily on ion transport proteins (Na⁺/H⁺ reverse transporter proteins and K⁺ channels) in the cell membrane to maintain the intracellular balance of Na⁺/K⁺ ratios, thereby prevent ion toxicity. In addition, salt-tolerant plants effectively scavenge excess reactive oxygen species (ROS) by increasing the activity of antioxidant enzymes (SOD, POD, CAT) and accumulating non-enzymatic antioxidants (APX, AAO, GR, and MDHAR), thereby protecting cells from oxidative damage. At the same time, plants increase cellular osmotic pressure and thus maintain the stability of cellular structure and water balance by accumulating osmoregulatory substances, such as Pro, SS and GB; phytohormones(including ABA, C2H4 and GA3) play a key role in regulating the plant's adaptation to saline and alkaline stresses, and they can enhance the plant's salinity tolerance by affecting the pathways of growth, stomatal opening and closing and the synthesis of osmoregulatory substances.

(2) Further studies also found that nano zinc oxide has a wide range of adaptability, it is not only suitable for crops, but also can be used for pasture, horticultural plants and many other types of plants, this wide range of applicability makes the nano zinc oxide in the field of plant protection has a broad application prospects, can be used to provide effective protection measures for different types of plants and agricultural production .
